# Interaction of Bacteria and Fleas, Focusing on the Plague Bacterium—A Review

**DOI:** 10.3390/microorganisms13112619

**Published:** 2025-11-18

**Authors:** Patric U. B. Vogel, Günter A. Schaub

**Affiliations:** 1Independent Researcher, 27476 Cuxhaven, Germany; patric.vogel@googlemail.com; 2Zoology/Parasitology, Ruhr-University Bochum, Universitätsstr. 150, 44780 Bochum, Germany

**Keywords:** antibacterial compounds, *Bartonella* sp., fleas, interactions, microbiota, plague, *Rickettsia* sp., *Yersinia pestis*

## Abstract

This review summarizes the interactions between three major bacterial groups, *Rickettsia* sp., *Bartonella* sp. and *Yersinia pestis*, the flea vectors and the diverse gut microbiota of fleas and highlights open questions. The focus is on the plague pathogen, *Y. pestis*, which adapted to transmission by fleas several thousand years ago. This caused one of the deadliest infectious diseases known to mankind, and the three pandemics resulted in an estimated 200 million deaths. In the vector, *Y. pestis* resists the adverse conditions, like other numerous bacterial species. *Rickettsia* sp. and *Bartonella* sp. as well as *Y. pestis* induce specific changes in the microbiota. The presence of bacteria in the ingested blood activates the production of antimicrobial proteins and reactive oxygen species, which normally have no effect on the development of *Y. pestis*. This bacterium infects mammals by different modes, first by an early-phase transmission and then by biofilm-mediated blockage of the foregut. Both interfere with blood ingestion and lead to reflux or regurgitation of intestinal contents containing *Y. pestis* into the bite site. Blockage of the gut leads to more attempts to ingest blood, increasing the risk of transmission. The lifespan of the fleas is also reduced. As *Y. pestis* is still endemic in wildlife in many regions of the world and human infections continue to occur in limited areas, studies of the interactions are needed to find new ways to control the disease.

## 1. Introduction

Bacteria of three genera are transmitted by fleas: *Rickettsia*, *Bartonella* and *Yersinia*, some of which cause important zoonotic diseases in humans [1,2,3]. Rickettsia *typhi* and *R. felis* are the causative agents of murine typhus and flea-borne spotted fever, respectively [4]. They occur worldwide, with the latter species often associated with the cat flea [5,6,7]. *Rickettsia* sp. are also widespread in ticks, mites and mosquitoes [5,8]. Although the incidence of human cases in the USA declined sharply in the mid-20th century due to vector control programs, it is currently recognized as an emerging cause of febrile illness, particularly in Africa [9,10,11].

*Bartonella* sp. are also distributed worldwide [12] and are the most common pathogen detected in fleas from dogs and cats in the UK [13]. Infection rates in wild small mammals are also generally very high [14,15]. In rodents, *Bartonella* sp. usually cause latent infections [12,16]. Several members of the genus *Bartonella*, including *B. henselae*, are zoonotic and can cause various diseases in humans [12,14,15,17]. For example, trench fever, which is characterized by recurrent episodes of fever and other symptoms such as headache and muscle pain, is caused by *B. quintana* and affected 1 million soldiers during the First World War [18,19]. There are several species of human-pathogenic *Bartonella* sp. that use fleas as vectors [20]. Of importance is the cat scratch disease pathogen, *B. henselae*, which is transmitted by cat fleas or by cats and is the most common *Bartonella* species causing disease in humans [15,16,19,21].

*Y. pestis* is the etiologic agent of plague. More than 350 mammal species are susceptible to infection with *Y. pestis* and can host the pathogen, especially wildlife species [22,23]. Humans are not the natural host. An early form of plague appears to have led to a sharp demographic decline in Neolithic farmers around 5000 cal. BP [24]. The plague went down in history as one of the deadliest infectious diseases known to mankind. Over the past two millennia, it has left an unprecedented footprint in human history. It has caused three pandemics: the first, known as the Justinian Plague, began in the 6th century, the second pandemic, which included the Black Death, began in the 14th century, and the third pandemic began in the late 19th century. Overall, plague caused an estimated 200 million deaths until the disease burden declined sharply in the 20th century [25]. Following the identification of the pathogen and its transmission, the decline coincided with improved standards or practices, including hygiene, housing and control of local rat populations [26]. In the 20th century, new inventions such as pesticides and antibiotics provided further tools to combat, control and treat plague. In the 21st century, infections in humans only occur sporadically, from single cases to local outbreaks [27,28,29,30], with about two thousand cases per year mainly in Africa [31]. It is still widespread but usually remains confined to so-called plague foci [30,32]. *Y. pestis* has been used for warfare in the past [33] and is a potent organism for bioterrorism [34].

In the wildlife, *Y. pestis* can cause outbreaks that are limited to individual animal families but also large epizootics that lead to a sudden decline in host populations [35,36,37,38,39]. These events are usually geographically limited and represent only the tip of the iceberg. The persistent disappearance and reappearance suggest sources of infection other than the typical mammal-flea-mammal cycle [40]. A “silent” enzootical cycle between the flea vectors and a variety of mammals, especially rodents, is controversially discussed but favored by the high persistence of the bacterium in different environments [33,36,41]. *Y. pestis* can survive either in the vector during winter or in association with hosts or reservoirs during hibernation [39,42]. Conserved genes are thought to be responsible for preservation after host death [43]. Outside the vector-host environment, *Y. pestis* can survive in soil for extended periods of several months [40,44,45]. In the environment, *Y. pestis* can tolerate high salinity through several outer membrane proteins, including an important efflux pump for passive transport of small hydrophilic molecules and Na^+^/H^+^ antiporters [46]. In addition, exposure to extreme or unfavorable conditions can also trigger stress responses to transform into a dormant L-shaped cell form [41].

Although *Y. pestis* is endemic in wildlife in many areas of the world, including North America, Africa and Asia, outbreaks and epizootics are rare among these mammalian hosts because many hosts have acquired varying degrees of resistance to *Y. pestis* [41]. One example is the circulation of *Y. pestis* in rat populations in natural endemic regions in Madagascar, which are orders of magnitude more resistant to plague than rats in other regions [47]. The immunological and immunogenetic basis for this resistance is only partially known [48]. Such continuous low-level circulation is one of the key factors that allow *Y. pestis* to survive in the environment. However, silent enzootic circulation leads to low bacterial concentrations in the host’s blood and may not be sufficient to ensure flea infection. Therefore, sporadic movement from fleas to susceptible hosts, resulting in high bacterial concentrations in individual hosts, is required to ensure long-term persistence within these plague foci [32]. Rodents can orally be infected, and when burrowing in contaminated soil. Oral transmission by infected animals in social communities living in close proximity or cannibalism may also contribute to spread during epizootics [49]. A vertical transmission, as proposed in a new concept, could also contribute to maintain *Y. pestis* in flea populations (Ref. [50]; see Section 5.2.3).

In addition to factors such as population size and susceptibility or resistance, the mode of transmission also influences plague dynamics in certain biomes [37,51]. Changes in ecosystems, e.g., climate changes, including temperature, precipitation and humidity, can affect either the host, the vector or the pathogen [52], which in turn affects the prevalence and burden of the disease. However, these influencing factors are multifactorial and complex, and responses cannot always be linked to single variables such as precipitation [53]. In China, population density and frequency of contact with reservoir animals in plague foci are among the risk factors [30]. In general, effects that lead to a decline in the natural host population increase the risk of a resurgence of plague in humans [33].

There is not a single transmission strategy for flea-borne diseases to mammalian hosts [2]. It varies and depends on the pathogen. *Rickettsia typhi* is transmitted through flea feces, whereas *R. felis* invades the salivary glands of cat fleas and is injected with saliva into the blood of cats, rodents and other hosts [6,54,55]. However, *R. felis* can also be transmitted horizontally between fleas during co-feeding on non-bacteremic hosts [56,57]. Additional transmission through feces is also possible, as *R. felis* DNA has been found in the feces of infected *Ctenocephalides felis* fleas up to four weeks after infection, with evidence of biologically active cells [11,58]. Therefore, the significance of the different transmission routes to mammals has yet to be established [11]. Transmission of * Bartonella henselae* occurs mainly directly through a cat bite or scratch and, less frequently, through cat flea feces [15,21], whereas *B. quintana* is excreted in the feces of body lice and rubbed into the wound [18].

*Y. pestis* is mainly transmitted by regurgitation of the contents of the flea’s anterior intestine into the wound [59,60,61,62] (see Section 6.1). Infections can also occur directly between humans, e.g., in pneumonic plague, where the pathogen enters the host’s lungs, multiplies there, and can be transmitted by droplets. However, the efficiency of this transmission route is considered low [41]. In addition, accidental infections can also occur through the processing or consumption of infected wildlife animals [33]. Another risk factor for transmission from wild animals to households is the presence of flea-infested rodents invading low-quality housing. For example, 40% of rodents from households in Yunnan Province, China, were infested with fleas [63].

Within the mammalian host, development and pathology vary depending on the flea-transmitted pathogen. *Rickettsia* are obligate intracellular bacteria [2,8]. *R. typhi* infects mammalian endothelial cells through clathrin-coated induction of phagocytosis and subsequent escape from the phagolysome. The resulting different clinical manifestations, known as murine typhus, begin with fever and may include, for example, headache, myalgia, but also rash or gastrointestinal symptoms [9,10,11]. The clinical symptoms of *R. felis* infections (flea-borne spotted fever) in humans are indistinguishable from murine typhus [11,64]. However, these rickettsial infections are successfully treated with antibiotics such as doxycycline [2,10]. In contrast, Rocky Mountain spotted fever, caused by *R. rickettsii* and transmitted by ticks, had a very high mortality rate before the use of antibiotics [9].

*Bartonella* infections in humans also lead to asymptomatic infections and severe disease [65]. After infection, *Bartonella* sp. are extracellular, disappear from the bloodstream, live facultatively intracellularly, e.g., in erythrocytes or endothelial cells, and thus hide from the immune system, and reappear in the bloodstream en masse a few days later, with up to 10^8^ viable cells per ml in cats, but without any noticeable symptoms [66]. Once they invade erythrocytes, they typically replicate at a low rate, infecting only a few erythrocytes and forming few daughter cells within them [16,66]. Most cases of cat scratch disease occur in children without serious effects, but immunocompromised individuals are affected, including chronic lymphadenopathy and other symptoms, while other *Bartonella* species often cause long-lasting, recurrent waves of bacteraemia [15,16]. Infection with *B. quintana*, the pathogen that causes trench fever, is chronic and can last for a maximum of several years [18].

Only a few *Y. pestis* cells are sufficient to infect highly susceptible hosts [67], while other mammals with higher levels of resistance require a significantly higher infectious dose [61]. They are taken up by immune cells such as macrophages and thus escape destruction. Through this initial intracellular stage, they acquire the ability to withstand further phagocytosis and switch to extracellular growth [68,69]. In susceptible hosts such as humans, lymph nodes enlarge and become painful [70]. *Y. pestis* spreads massively throughout the body, which contributes to its lethal effect, particularly after development in the lungs [71]. Systemic spread from peripheral sites is facilitated by the plasmid-encoded virulence factor, a plasminogen activator [41,69]. The early stages of infection are successfully treated with antibiotics [2]. There is no vaccination against *Rickettsia* sp. and *Bartonella* sp., and vaccines against plague are in limited use but are neither licensed nor commercially available in most countries [72,73,74].

The life cycles of *Rickettsia* spp., *Bartonella* spp. and *Y. pestis* are characterized by a continuous alternation between two distinct environments, the mammalian host and the flea vector, which differ in many physiological aspects [62,75]. However, variations can be observed. *Rickettsia* sp. utilize different strategies, ranging from entirely vertical transmission to a mixture of horizontal and vertical transmission [8]. *R. felis* is highly adapted to the non-hematophagous booklouse, that transmits it to humans and is fully maintained by vertical transmission; in addition to the aforementioned host-vector switching, the infection may also be maintained horizontally [56,57]. In fleas, *R. felis* develops in the intestinal tract and infects the salivary glands [4]. Different *Bartonella* spp. vary in their location inside the flea vector [20,21] (see Section 5.2.1).

After ingestion by fleas, *Y. pestis* also remains primarily extracellularly localized in the foregut and midgut [62] (see Section 5.2.3) (Figure 1). In the gut, the bacterium is exposed to lysed blood components, digestive enzymes, humoral immune effectors of the flea and the natural commensal and pathogenic microbiota. It must adapt to this microenvironment and manipulate the vector to maintain its life cycle. *Y. pestis* faces several major challenges in the flea midgut: (1) defense against flea antimicrobial responses, (2) metabolic adaptation to continue proliferation despite fundamental changes in nutrient supply, (3) competition with other microbiota for colonization of the flea vector, (4) stabilization in the proventriculus at the end of the foregut by aggregation and subsequent biofilm formation to prevent excretion from the flea gut through defecation and (5) interference with blood-feeding through foregut obstruction, including biofilm expansion, which narrows or blocks the lumen of the proventriculus and results in a regurgitation of bacteria into the bite site of the flea vector [59,61,62,76,77].

The number of annual scientific publications on *Y. pestis* has decreased over the last decade, but profound insights into the evolutionary development and occurrence [24,78,79] as well as molecular interactions with the vector and the course of infection in fleas have been gained in the past years. For example, the principles and relative efficiency of the main transmission modes, as well as other factors influencing the course of infection, such as blood sources, the colonization process, but also the initial progression of disease in rodents, have been further substantiated by controlled study designs, including fluorescence-based analyses depicting early and late events on a cellular basis [50,51,80,81,82,83,84,85,86]. A thorough understanding of pathogen-vector interactions is fundamental to understand the mechanisms that influence vector specificity, transmission and persistence in plague outbreaks, identifying new approaches to vector control and predicting the epidemiological pattern of plague. This is particularly important as climate change progresses with difficult-to-predict consequences for plague resurgence, increasing pesticide resistance in flea populations and a lack of effective prevention measures [74,87,88,89,90]. Therefore, a better understanding of the ecological, biological and molecular aspects of pathogen-vector interactions is essential to prevent and control plague.

## 2. *Rickettsia*, *Bartonella* and *Yersinia*

These three genera all belong to Proteobacteria, Gram-negative coccobacilli [6,12,17,91,92]. *Rickettsia* sp. and *Bartonella* sp. belong to the class Alphaproteobacteria [93]. The genus *Rickettsia* comprises > 30 species. These bacteria are characterized by an obligate intracellular lifestyle [8]. Three phylogenetic clades are distinguished: the typhus group (e.g., *R. typhus*, *R. prowazekii*), which uses fleas and lice as vectors; the spotted fever group (e.g., *R. rickettsii*), which uses ticks as vectors; and the transitional group (e.g., *R. felis*), which uses a variety of different arthropods as vectors. In addition to well-characterized species, there is an increasing number of *R. felis*-like species worldwide, genetic variants that have been detected in fleas and are likely to be classified as new *Rickettsia* species [8,10,11]. The different groups of *Rickettsia* sp. exhibit considerable genetic differences, including genome size, the presence of mobile genetic elements, gene rearrangements, etc. [9].

The genus *Bartonella* comprises > 80 species and subspecies [14]. Of these, >15 can cause human diseases [15], with *B. bacilliformis* being the most pathogenic species [16]. Within the genus, few lineages are distinguished [14,16]. *Bartonella* sp. can infect a wide variety of mammals, but rodents in particular are natural reservoirs and important for the persistence of *Bartonella* [12,14]. Only two, *B. bacilliformis* and *B. quintana*, naturally use humans as a reservoir [16]. *Bartonella* sp. are facultatively intracellular and prefer two different cell types, endothelial cells and erythrocytes, thus hiding from the immune system [12,16]. *Bartonella* sp. also possess various virulence factors to evade the immune response of their mammalian hosts [94].

*Y. pestis*, which often exhibits bipolar staining in Giemsa staining [34,67,95], belongs to the class Gammaproteobacteria, which also includes other pathogenic bacteria such as *Vibrio cholerae*, the causative agent of cholera [96]. *Y. pestis* belongs to the order Enterobacterales [97], which includes numerous non-pathogenic and pathogenic bacteria that live in the gastrointestinal tract of animals and humans, such as the genera *Shigella* and *Salmonella* [98]. The genus *Yersinia* consists of >25 species and includes non-pathogenic and three primarily pathogenic species: *Y. pestis*, *Y. enterocolitica* and *Y. pseudotuberculosis* [69,98,99,100,101]. The latter two bacteria, like the aforementioned Enterobacterales, are associated with the gastrointestinal tract and are transmitted from person to person through feces, contaminated water or food. They primarily cause gastrointestinal diseases, known as yersiniosis. In the EU, yersiniosis caused by these species has been the third most common intestinal disease in humans in recent years, particularly among young children [102,103].

*Y. pseudotuberculosis* is the direct ancestor of *Y. pestis*, from which it diverged approximately 6000 to 8000 years ago [26,72,79,104]. *Y. pestis* is the only species of the genus that has adapted to insect vectors. About 4000 years ago, a permanent cycle may have developed between some mammals such as brown rats and fleas, after *Y. pestis* acquired the relevant genetical adaptations for transmission by fleas [78,85]. *Y. pestis* possesses 4.65 Mb of chromosomal DNA encoding approximately 4000 genes [105]. The pathogen retained a conserved chromosomal DNA with high intraspecific sequence similarity, but also high similarity to its predecessor [106,107], although the extrachromosomal elements exhibit significantly higher genetic diversity [108].

Overall, genetic adaptations included gene losses or silencing, as well as the acquisition of plasmids encoding various virulence factors [72,77,109,110]. *Y. pestis* harbors three different plasmids with plasmid-specific copy numbers ranging from a few to >100 per cell [105,111]. Of particular importance are a few mutations that enable *Y. pestis* to produce transmissible infections in fleas. These include the acquisition of the plasmid-encoded yersinia murine toxin gene, three loss-of-function mutations involved in biofilm formation and the loss of urease activity, which has lethal effects on fleas [104,109,110]. Particularly the yersinia murine toxin contributed to or enabled a broad host range [85,112] (see Section 5.2.3). However, this is only a general summary, as the molecular processes and interactions are significantly more complex; for example, many gene products are required for proventricular blockage alone [61,62]. Dozens of chromosomal genes of *Y. pestis* are missing in *Y. pseudotuberculosis*, and over 200 genes of *Y. pseudotuberculosis* have become nonfunctional pseudogenes of *Y. pestis* due to metabolic adaptation to the new lifestyle [59,77,105,113]. A specific two-component signal transduction system, the PhoP-PhoQ-System, consisting of a regulator and a membrane sensor, plays a crucial role in the adaptation of *Y. pestis* to the vector and the host. Loss-of-function mutants show impaired survival in immune cells in vitro, as well as reduced biofilm formation and blockage of flea vectors, which negatively impacts transmission efficiency [114,115]. Of the approximately 30 two-component systems encoded in the *Y. pestis* genome, at least two others are important for adaptation to the flea’s intestinal environment in response to environmental stimuli [62,115,116,117] (see Section 5.2.2 and Section 6.2).

*Y. pestis* and *Y. pseudotuberculosis* are genetically substantially distinct from *Y. enterocolitica* [97], which have been isolated from a wide range of animal hosts [99], with the latter two species being motile at moderate growth temperatures [100]. All three pathogenic species share some virulence factors, including the siderophore yersiniabactin, which is essential for iron uptake [69,98,100]. It is encoded in the so-called “high pathogenicity island”, which is part of the chromosomal DNA, and the iron uptake system is essential for in vivo growth [118]. While the mammalian host attempts to sequester metal ions such as iron and zinc to limit microbial growth (so-called nutritional immunity), *Y. pestis* overcomes this metabolic limitation by secreting yersiniabactin [119,120]. *Yersinia* can manipulate the host’s immune response, for example, through a plasmid-encoded type III secretion system that allows the injection of outer membrane proteins into target cells to disrupt immune signaling and ultimately inhibit phagocytosis [97,98]. A similar strategy using a type IV secretion system to release effector proteins to disrupt host cell apoptosis allows *Bartonella* sp. to persist in erythrocytes of mammalian hosts [94].

With few exceptions, members of the genus *Yersinia* also share other common biochemical properties, such as being catalase-positive, oxidase-negative and capable of glucose fermentation. Based on their biochemical properties, they can be identified using Analytical Profile Index (API) 20E test strips [98,100]. The most important marker for laboratory detection of *Y. pestis* is the F1 capsular antigen [95], which is highly transcriptionally upregulated in vitro at 37 °C [121]. It is considered an important adaptation of *Y. pestis* to achieve high bacteremia in mammalian hosts, facilitating flea infection and a continuous host-vector cycle [122]. The pathogen lacks the O antigen, a typical virulence factor of other enteric bacteria [123], including *Y. enterocolitica* [97]. This appears to be another important adaptation that allows *Y. pestis* to invade dendritic cells, since O-antigen-positive mutants are unable to disseminate from the initial inoculation site to the lymph nodes in mouse infections [124]. Some *Y. pestis* strains exhibit biochemical differences that form the basis for biovar differentiation [107]. Historically, each of the three human pandemics was caused by different biovars [33]. However, analysis of ancient samples also suggests that one of the biovars, Orientalis, may have been present during all three pandemics [69].

*Y. pestis* exhibits some unique metabolic limitations that make it an obligate parasite with both aerobic and facultative anaerobic metabolism. For example, it lacks several metabolic enzymes and requires biotin, thiamine and glutamic acid for growth at elevated temperatures [67,125]. In vitro, the bacterium grows slowly, with doubling times on standard media of approximately 1.25 to 2 h [67,125] and requires the addition of several amino acids to the culture medium [34]. Therefore, yersiniae are often overgrown by other undemanding bacteria in mixed samples [98], which can also occur in experimental studies investigating pathogen-vector interactions [126,127]. For experimental purposes such as flea infections with artificial systems, pure stock cultures of *Y. pestis* typically grow well in Brain Heart Infusion Medium [49,51,61,80,128]. *Y. pestis* grows rapidly in the flea vector [62]. However, the in vivo growth rate does not appear to be the main factor for vector colonization. For example, *Bartonella* species that can infect fleas require about one day for cell division [129], which is atypical for bacteria and very close to the doubling time of eukaryotic cells [101].

*Y. pestis* has a pH optimum for growth of approximately 7.2 to 7.6 but also tolerates a wider pH range [67]. This is well adapted to the slightly more acidic pH value in the midgut of fleas [62] but also to the blood of mammals.

## 3. Fleas

### 3.1. Occurrence, Morphology and Development

Fleas are distributed worldwide and even parasitize birds in Antarctica [130]. There are over 2000 flea species [3,131,132]. Of these, about three-quarters are ectoparasites of rodent species and only a few live synanthropically with humans [2]. Fleas mainly infest hosts of wild and domestic mammals, e.g., rodents, dogs and cats that live or rest in permanent sleeping places or burrows, while mammals with changing resting places usually do not have infestations [2]. Adult fleas live near their hosts, either permanently on them or in their nests [133,134]. The flea infestation rate in companion animals depends on several factors but commonly ranges between 10% and 40%, with the infestation rate in dogs more often below 10% [13].

Adult fleas can be easily recognized by the following morphological characteristics: The wingless, laterally flattened flea body, measuring approximately 1 to 6 mm, is heavily sclerotized and has posteriorly directed spines on various parts of the body that assist movement and prevent easy dislodging [2,3]. The antennae are very short [135]. Populations of the human flea *Pulex irritans* exhibit high morphological conservation and low genetic variation across different geographical regions [136,137]. The extremely powerful hind legs enable them to cover great distances, with average jumps of 20 to 30 cm in *Ctenocephalides* species and jumping speeds of 1.4 to 3.6 m/s in the black rat flea *Xenopsylla cheopsis* and the cat flea *C. felis* [138,139].

Fleas belong taxonomically to the order Siphonaptera in the class Insecta and are holometabolous insects characterized by four developmental stages: eggs, larvae, pupae and adults [135]. The number of eggs laid per day depends on the species. While some species lay about 5 eggs per day [50], adult females of *C. felis* lay up to 40 eggs per day, about 1700 eggs in their lifetime and each 0.3 to 0.5 mm long [139,140]. The eggs are laid in the host’s fur and surrounding areas [2]. Embryonic development takes about 1 to 10 days [141]. After hatching, the eyeless flea larvae move actively and, depending on the larval stage, feed on organic detritus, other eggs and droplets of adult flea feces containing partially digested or digested blood remnants [2]. *C. felis* larvae that additionally feed on eggs have a strong developmental advantage: 90% become adults, whereas only 10% of larvae that feed on feces but not eggs successfully develop into adults [140]. After three larval stages, they are 4 to 10 mm long [134]. They then spin a pupal cocoon in which they develop to adults, i.e., larval tissue is lysed (histolysis) and adult tissue is formed (histogenesis). Not all, but some flea species require a stimulus such as vibrations from the host to induce the emergence of adult fleas from the pupal cuticle [2]. Full development until the adults hatch depends on temperature and humidity but takes three to five weeks on average [133], although in the case of the rat flea, under optimal conditions, it can take only about 1.5 to 2 weeks.

### 3.2. Host Preferences

Some flea species show a preference for specific hosts or host groups, such as typical indigenous species/subspecies in Australia [142]. Furthermore, the association between the rabbit flea *Spilopsyllus cuniculi* and the European rabbit is very strong, and host hormone levels influence the flea’s reproductive cycle [143]. In contrast, many flea species are undemanding and exploit a wide range of mammalian hosts, such as the cat flea and the human flea [3,133,144]. The latter is associated with humans but is mainly found on foxes and badgers and occasionally parasitizes many other hosts, e.g., pigs and other large mammals [143].

Once established on a host, fleas such as *C. felis* have little tendency to switch hosts [140]. While this occurs with direct contact with cats, the tendency decreases sharply after fleas habituate individual cats for two days [145]. Fleas typically remain on the host for >100 days, but countermeasures such as grooming can significantly reduce this time to less than 10 days in cat fleas [140]. In other flea-host pairs, such as *Xenopsylla cheopis* and rat species, host switching appears to be more frequent [146]. Switching to new hosts does not necessarily lead to behavioral adaptations and may be species-dependent. No adaptation in feeding performance is observed in two flea species when they were kept on new hosts for 15 generations. However, other flea species show increasing specialization on the new host while losing feeding performance on the natural host [147,148,149].

### 3.3. Attraction and Blood Ingestion and Digestion

Adult fleas are obligate bloodsucking insects [134] (Figure 2). The newly emerged adult flea immediately seeks a host. Various host factors such as carbon dioxide, movement and body heat serve as attractants [2]. Once they have found a host, they immediately begin feeding and mate. There are significant differences in the frequency of feeding between flea species, with *X. skrjabini* and *C. felis* feeding daily or several times a day, while *X. cheopis* and *O. montana* feed less frequently [35,61,128,150]. Cat fleas require 10 to 25 min to ingest blood, and most fleas have finished feeding after one hour [140]. Adult fleas ingest approximately 0.3 µL blood during a single feeding, with female *X. cheopis* fleas ingesting on average twice as much blood as males [85,151]. During this process, the fleas inject their saliva into the bite site. In other arthropod vectors, the saliva contains a variety of pharmacological agents involved in hemostasis, immune modulation and pathogen transmission [152,153]. Compared to other blood-feeding arthropods, flea saliva is less complex, but its composition, components and protein families that may act as effector molecules are increasingly being analyzed [154,155,156,157].

Blood uptake occurs through alternating contractions of two muscle-driven pumps, the cibarial and pharyngeal pumps, which produce a directed blood flow along the digestive tract [126]. The blood flows through the foregut, i.e., the esophagus and the proventriculus, a valve open to the midgut at the junction of the foregut and midgut, which opens and closes rhythmically during blood uptake [77,126,143]. The proventriculus contains inner rows of spines that presumably lyse erythrocytes. Its pulsations and peristaltic contractions of the midgut mix the blood with digestive enzymes [62]. The hemolysis process is rapid, occurring within a few hours, compared to other insect vectors such as mosquitoes and triatomines [158,159], and hemolyzed blood remains liquefied in the midgut [77]. The blood source clearly influences the speed of the fleas’ blood digestion. Mouse blood is rapidly hemolyzed, whereas in *X. cheopis*, which is fed on blood from brown rats, intact red blood cells can still be seen in the midgut contents eight hours after blood ingestion [85]. Generally, blood is stored and digested throughout the whole midgut before the remnants reach the hindgut [143]. After a single feeding of *Xenopsylla ramesis* on male or female rodents, the gender of the host influences the rate of blood digestion in the middle and late phases [160]. In this context, the progress of blood digestion correlates with the ability to infect fleas and sex differences in flea infection rates, with lower infection rates in male fleas, have been observed for some blood sources [85]. Therefore, the influence of gender-specific digestive kinetics on *Y. pestis* infection should be further investigated.

In response to blood ingestion, fleas upregulate various genes, including those of serine proteases, which are the most abundant digestive enzymes in the flea midgut [161]. These flea digestive enzymes are similar to those found in most other blood-feeding insects, trypsins and chymotrypsins [83]. After one day, fleas excrete digested and partially digested blood in droplets [62].

### 3.4. Conditions in the Intestinal Tract of Fleas

The pH in the flea midgut is slightly more acidic than in the blood, but the osmolarity is similar. After blood ingestion, the pH appears to rise from 6.6 to 7.0 and returns to its original pH after 24 h. The osmolarity of 300 mOsm in unfed fleas immediately increases significantly to 500 mOsm upon blood ingestion and also returns to its original value 24 h later [116]. The flea midgut is surrounded by an extensive abdominal tracheal system and appears to be well oxygenated, as indicated by the brightness of oxygen-dependent green fluorescent protein molecules in the digestive tract [62]. The activity of the tracheal system of fleas increases during blood digestion [162].

Many insects produce peritrophic membranes that enclose blood contents, including potentially pathogenic microorganisms, to prevent damage or infection of midgut epithelial cells [163]. These membranes are composed of proteins, primarily peritrophins and chitin, secreted by midgut epithelial cells and cells of the cardia, the valve at the junction of the foregut and midgut that prevents the backflow of ingested blood [164]. In fleas, the cardia is commonly referred to as the proventriculus [143]. Unlike other insect vectors, fleas do not form protective peritrophic membranes but synthesize various peritrophin-like proteins. These are also present in unfed fleas and therefore do not require blood ingestion for synthesis [158,165]. Peritrophins are not found in the midgut, but rather in the Malpighian tubules, hindgut, rectum and trachea of the cat flea [166,167]. Some are upregulated in the presence of *Y. pestis* in the blood and they presumably have various biological functions, such as the reabsorption of ions and nutrients, the coating of spines in the proventriculus or tracheal repair [83,166].

### 3.5. Against Pathogens: The Immune System of Fleas

A general defense strategy of all animals against microbiota and pathogens is an immune response, which is also elicited by many invertebrates. Pathogen detection is based on the recognition of pathogen-associated molecular patterns characteristic of specific pathogens by corresponding receptors [168,169]. For example, lipopolysaccharides from the outer cell membrane of Gram-negative bacteria are potent pathogen-associated molecular patterns that are also recognized by mammalian and insect vectors such as triatomines [170,171]. A peptidoglycan recognition protein is upregulated in the cat flea after blood ingestion, inducing an enzymatic degradation of peptidoglycans from bacterial cell walls [161]. These molecular patterns activate the immune system, trigger various immune signaling cascades and induce the synthesis of immune effectors [168]. In the model insect *Drosophila*, the immune response is activated via five signaling pathways: Toll, immunodeficiency (Imd), Janus kinase/signal transducer and activator of transcription (JAK/STAT), Jun N-terminal kinase and mitogen-activated protein kinase [169]. The Toll signaling pathway is activated against Gram-positive bacteria and fungi, while the Imd signaling pathway responds to Gram-negative and some Gram-positive bacteria. The JAK/STAT signaling pathway is activated by viruses [169]. However, not all insects, such as mosquitoes, distinguish between Gram-positive and Gram-negative bacteria, and cross-induction can occur, as in the triatomine *Rhodnius prolixus* [172,173].

So far, only all genes required for Toll and Imd have been identified in fleas [174]. In the cat flea, the antimicrobial activity in the midgut is increased by bacteria [129]. In these fleas, the Toll signaling pathway is not stimulated during experimental infections with selected Gram-positive and Gram-negative bacteria. However, both bacteria (*Micrococcus luteus* and *Serratia marcescens*) activate components of the Imd signaling pathway in a species-dependent manner. Interestingly, *B. henselae* does not activate the Imd signaling pathway [129]. Specific memory during subsequent infections of insects with bacteria [175] has not been studied in fleas.

The immune system of arthropods, and thus also of fleas, consists of two parts with different effector functions: a cellular and a humoral response [168,169]. The cellular part of the immune system of fleas is far less investigated, but, like other arthropods, includes various types of hemocytes [169]. The humoral response leads to the synthesis of soluble immune effectors. Two of these immune effectors, reactive oxygen species and antimicrobial peptides (AMPs), are used by fleas to combat microorganisms [176]. In *Drosophila*, these oxygen species are produced by intestinal epithelial cells independently of the Imd signaling pathway [177]. AMPs are widely distributed throughout the animal kingdom, from simple multicellular organisms to mammals, and in some insects, many different AMPs and many isoenzymes of some AMPs are synthesized [83,168,169,178]. Fleas produce AMPs in response to blood ingestion and microorganisms [83]. These include defensin-2, two attacins and a coleoptericin-like peptide [169]. The cells of the proventriculus secrete these AMPs, which then diffuse into the midgut [179].

## 4. The Microbiota of Fleas

### 4.1. General Aspects of Insect Microbiota

Like all organisms with a digestive tract, blood-feeding arthropods host a microbial community in their gut [178]. Some harbor microorganisms in specialized cells and/or in association with internal tissues. Offspring acquire the microbiota through various strategies. Some are transferred transovarially from adults to offspring through incorporation into the egg cell and then develop intracellularly, but most are acquired through active uptake during early stages. For example, nymphs of triatomines, the vectors of Chagas disease, contact the feces of other triatomines and thus acquire the typical gut microbiota, including the mutualistic symbiont [153]. New generations of mosquitoes also largely obtain their gut microbiota from the environment [180]. Microorganisms can also be found in the blood of hosts such as rodents [181]. These bacteria are then ingested during blood ingestion.

There is no uniform microbiome in terms of abundance and composition for a particular vector species, and the intestinal flora varies considerably between regions and populations [178,180,182,183,184]. The gut microbiota are often commensals but can support the insect vector in various ways, including nutrient supply, detoxification, digestive support and prevention of pathogen colonization [185], often acting as mutualistic symbionts. The term “symbiont” is not used below because the mutualistic symbionts of fleas have not been identified.

### 4.2. Microbiota of Fleas: Infection and Groups of Bacteria

Flea larvae acquire microorganisms from the environment and from the feces of the flea population (Ref. [176], see Section 3.1). Adults come into contact with the skin of their hosts and can ingest infectious blood. The full microbial spectrum of the flea digestive tract has not been studied in detail, as many studies use whole flea bodies for DNA isolation. Fleas exhibit a high diversity of microbiota, with the microbial flora differing between individuals, populations, species and over time [147,186,187]. Next-generation sequencing has identified >1800 species in the gut of the cat flea *C. felis*, with the majority (>1400) of the organisms being bacteria [188]. Individual fleas often host multiple strains or many species of the bacterial groups [189].

The microorganisms most commonly associated with flea bodies belong to the phyla Proteobacteria and Bacteroidetes and, depending on the region and flea species, to the phyla Firmicutes (taxonomically renamed Bacillota), Actinobacteria and Tenericutes [142,186,190]. Firmicutes and Bacteroidetes are among the most abundant gut microbiota of a wide variety of animals, including humans, and Proteobacteria and Firmicutes also represent the most abundant microbiota of insects including mosquitoes [180,182,191,192], despite great variation in breeding sites (dry nesting sites, soil, or standing water). The Gram-positive Firmicutes commensals have beneficial effects on the host by supporting nutrient metabolism, preventing colonization by pathogenic bacteria and stabilizing intestinal barrier function [191]. Firmicutes include representatives of various genera, such as *Bacillus* and *Staphylococcus*, and contain both aerobic and anaerobic bacteria [193,194]. In contrast to the human digestive tract, which is dominated by anaerobic bacteria [195], the flea midgut appears to be well oxygenated and thus supports colonization by aerobic bacteria (see Section 3.4).

An important aspect of studying flea microbiota is its maintenance in the laboratory. The flea microbiota changes slightly when fleas are acclimated to laboratory conditions. While Proteobacteria and Actinobacteriodes remain unaffected, members of the Bacteroidetes disappear under laboratory conditions [186]. Some laboratory colonies are maintained for 10 to 25 years [196]. These colonies may have either no or a significant altered microbiota, as many factors, such as host factors including blood composition, skin flora, but also the environment can influence the microbiota. The influence of laboratory conditions, including the continuous adaptation of laboratory strains, the absence of natural environmental stimuli, sources of contamination, etc., strongly influences the microbiota in many parasite-vector systems [153,178]. In the case of *Yersinia*, exposure of laboratory fleas to wild flea feces and soil from natural wildlife burrows of black-tailed prairie dogs increased subsequent transmission rates of *Y. pestis* [197].

The flea gut is colonized by many Gram-negative bacteria [176]. Some are found less frequently, e.g., Gram-negative *Cardinium* sp. which was found in one *Xenopsylla* species and Gram-negative *Achromobacter* and Gram-positive *Rhodococcus* in *C. felis* [189]. The cat flea also harbors members of the two Gram-positive families Lachnospiraceae and Ruminococcaceae, which are known to support digestion in mammals and other insects [198]. Technical advances such as shotgun sequencing, which is used instead of selective testing for specific pathogens using PCR, have also revealed many new potential microbiota species, but on the genus level, *Rickettsia*, *Bartonella* and *Wolbachia* species often dominate the flea microbiota, e.g., in wild cat fleas [188,198,199].

#### 4.2.1. *Rickettsia* sp.

Fleas acquire these bacteria orally through blood ingestion from infected hosts, leading to stable infection [58,200]. A second mode of transmission to other fleas occurs through mechanical contamination of the mouthparts and release during probing or through the skin during prolonged feeding on a host [5,11,56,201]. This horizontal transmission from an infected to an uninfected flea by feeding on the same host also occurs, for example, between cat fleas or *Xenopsylla cheopis*, or other insect vectors such as mosquitoes [56,202]. Fleas can transmit *R. felis* through co-feeding within 24 h of infection without the pathogen spreading within the flea body, provided the co-feeding lasts at least 12 h [56]. *R. felis* can also be transmitted vertically by cat fleas and can be maintained in the flea population for >10 generations, although infection rates decrease [58]. In other arthropods, *R. felis* colonizes the vector either transiently (e.g., mosquitoes) or for the duration of the infected generation (e.g., ticks) [11]. In tick cell lines, intracellular growth of *R. parkeri* induces apoptosis, which promotes further cell infection by the pathogen rather than clearing the infection [203]. However, it is unknown whether this cellular mechanism plays a role in the interaction between *Rickettsia* and fleas.

#### 4.2.2. *Bartonella* sp.

The genus *Bartonella* is also commonly found in fleas, particularly the cat flea *C. felis*, but also in other flea species and insect vectors such as lice and ticks [14,15,17,186,198]. *Bartonella* is highly adapted to its flea vector [204]. In addition to transmission through flea feces, *Bartonella* can also be transmitted horizontally and vertically within flea and rodent populations, leading to high prevalence rates in rodent populations [12]. However, *Bartonella* infection of rodents does not necessarily correlate with flea infestation, as observed in brown rats in Canada [205]. It still seems to be the most frequently detected microorganism in fleas, and certain *Bartonella* species dominate in cat flea populations in certain geographical regions [186,190,206].

#### 4.2.3. *Wolbachia* sp.

*Wolbachia* sp. are obligate intracellular bacteria of invertebrates and, like *Rickettsia* spp., belong to the Rickettsiaceae family [207,208]. *Wolbachia* are among the most commonly observed endosymbionts in nature and inhabit many invertebrates [209]. *Wolbachia* species occur in various arthropod vectors, including triatomines and tsetse flies [178,210]. In these insects, *Wolbachia* multiply in the reproductive system and can cause reproductive disorders [153]. Unlike other bacteria, they are not ingested orally from the environment but are transmitted vertically from adult female fleas to their offspring. This facilitates long-term coexistence with their hosts and enables continuous adaptation. This host adaptation can also be facilitated by lateral gene transfer. The genus *Wolbachia* is generally susceptible to this form of gene acquisition, as has been demonstrated in other invertebrate hosts and also in fleas [211,212]. For some flea species and regions, *Wolbachia* is the dominant bacterium [140,142]. The endobacterium prefers female fleas, which have the highest infestation prevalence at approximately 70% [209,213]. The presence of multiple *Wolbachia* lineages in cat fleas is attributed to laboratory breeding [198] and may not be representative of the natural environment. Nevertheless, one lineage usually dominates numerically in individual fleas [214]. Despite their long coexistence, new strains and species are still frequently discovered in fleas, whose taxonomic relationship within the genus *Wolbachia* is dynamically redefined [199,211,215,216,217].

*Wolbachia* sp. require metabolites from their hosts to complement their own reduced metabolism [211]. However, this strong association with female fleas in particular does not represent a mutualistic symbiosis, as fleas cleared from *Wolbachia* sp. show no recognizable negative effects, not even on reproduction rate and fitness [209]. In other insects such as leafhoppers or mosquitoes, endosymbiotic *Wolbachia* are known to downregulate overall microbiota diversity or at least certain gut bacteria [218,219]. However, it is unknown whether *Wolbachia* has a similar effect in fleas.

#### 4.2.4. Interactions of *Rickettsia* sp., *Bartonella* sp., *Yersinia pestis* and Other Bacteria in the Flea

*R. felis* reduced microbiota diversity in two of three cat flea colonies [220]. Interactions also occur between *Rickettsia* spp. in coinfections of cat fleas with *R. felis* and *R. typhi*. While *R. typhi* benefits in terms of infection load from prior exposure to *R. felis*, the presence of *R. typhi* reduces the growth of *R. felis*. However, the loads of both species in flea feces are higher two weeks after infection than in single infections. This suggests that coinfection can enhance horizontal transmission [64]. *Bartonella* and *Wolbachia* levels are unaffected by *Yersinia* infection of rock squirrel fleas [186]. These two therefore either do not compete with *Yersinia*, are more resistant to the humoral immunity elicited by the flea (*Bartonella* is Gram-negative, as is *Yersinia*), or they are otherwise protected, e.g., by physical separation, as *Wolbachia* develops intracellularly in different tissues but not freely in the midgut. *Bartonella* may also not be in direct spatial competition with *Yersinia*, as it has been found in the midgut, hemolymph and ovaries, or it may be protected due to its potential ability to form biofilms in fleas [20,221]. For example, *Bartonella brasiliensis*, another human pathogen, colonizes the midgut of the sand fly vector *Lutzomyia verrucarum* between the peritrophic membranes and the endothelial cells [222]. Furthermore, sand flies experimentally infected with *Bartonella ancashensis* are transiently colonized in the anterior midgut and display aggregated bacteria organized into microcolonies [223]. Feeding the cat flea *C. felis* on *Bartonella henselae*-infected cats, microbiota diversity increases 24 h after infection and then decreases after nine days to levels comparable to those of unfed and uninfected controls. However, the underlying mechanisms are still unknown [198].

Little is known about how the normal gut microbiota of fleas influences *Y. pestis* and vector competence [77]. In *Y. pestis* infections of rock squirrel fleas, *Bartonella* and *Wolbachia* remain unaffected (see above), but other microbial diversity, e.g., the Firmicutes population, is markedly reduced [186]. Compared to Gram-negative bacteria such as *Yersinia*, Gram-positive bacteria are on average more sensitive to antimicrobials [224] and may also be less resistant to toxic or damaging components such as AMPs or reactive oxygen species. However, further research is needed to determine whether this decline is a permanent consequence of *Yersinia* infection and to analyze the underlying mechanisms. It is not known whether this is mediated by specific effector molecules, competition for nutrients or the ability to multiply rapidly and thus overgrow more fastidious and slow-growing bacteria, or an induction of flea immune defenses.

An interaction is evident in other parasite-vector systems: the midgut microbiota of the sand fly *Lutzomyia longipalpis* is crucial for the development of *Leishmania infantum*, and the absence of this microbiota impairs parasite growth and metacyclogenesis, resulting in sand flies unable to promote parasite transmission [225]. *Trypanosoma cruzi* specifically induces immunity of the vector triatomine, thereby reducing the populations of those bacteria that would otherwise impair the trypanosome [178].

In summary, considering that fleas can host over 1800 microbial species in their gut [188], the interactions with and effects on specific microbiota can be complex and current knowledge may only scratch the surface.

## 5. Interactions of Fleas with the Bacteria

### 5.1. Associations of Fleas and Bacteria

Cat fleas transmit various pathogens, including *Rickettsia* and *Bartonella* [140]. The rat flea, *X. cheopis*, and rats are the primary vector-host pair that occasionally leads to human rickettsiosis (*R. typhi*), particularly in warm coastal areas and port cities in the tropics and subtropics, but the cat flea *C. felis* is also a primary vector in some regions [6,9,10,174,215].

*Y. pestis* has been found in >250 flea species in the field and several dozen appear to be true vectors [22,226]. General differences between different flea species in their ability to transmit *Y. pestis*, regardless of the mode of transmission, were demonstrated many decades ago [227]. *Xenopsylla* and *Oropsylla* show a high transmission efficiency [196,228] and *X. cheopis* is considered the main vector for plague (although it is not necessarily the main cause of plague in all geographical regions [41,128,129,229]. In contrast, the synanthropic cat flea *C. felis* is a competent vector of *Y. pestis* but shows a low transmission efficiency, as a high midgut turnover leads to an accelerated defecation [128,230]. However, cat fleas are the most common ectoparasites of dogs and cats and thus provide a link between domestic animals, wildlife and human households [13], which could compensate for low efficiency with high infestation rates.

The anatomy of the foregut of different flea species appears to have a major influence on their ability to transmit *Y. pestis*. For example, the cat flea has a thick muscle layer at the esophagus-proventriculus intersection, possibly limiting biomass extension, whereas other flea species with a thinner muscle layer, such as *X. cheopis*, exhibit greater biofilm extrusion from the densely colonized proventriculus into the esophagus. The ratio of esophageal to proventriculus width also differs between species [128,196].

The human flea *P. irritans* is assumed to be a plague vector, as fleas infected with *Y. pestis* have been found predominantly in households within the vector during outbreaks [231,232]. However, this ability depends on host factors including blood source and is particularly low when fed on human blood that does not induce foregut blockage [49]. Furthermore, the ability to transmit *Y. pestis* in the first few days after infection is low [61]. In summary, a significant contribution to plague epidemiology is unlikely [233]. These findings contradict the common view that human ectoparasites including *P. irritans* are relevant vectors during plague epidemics [234]. However, an important role should not be excluded due to *P. irritans*’ ability to rapidly excrete viable *Y. pestis* cells, which represents a source of infection [41], and due to poor hygiene and living standards as well as high infestation rates in the human population at the time of plague epidemics, which may have overcompensated low transmission efficiency. In addition to fleas, the human body louse has also been confirmed as a vector for *Y. pestis* [235]. This ectoparasite also lives on humans and is an obligate daily blood feeder [236]. Originally, it was assumed that the body louse transmits *Y. pestis* exclusively via feces, which contain a large number of viable cells, as the body louse lacks the typical arthropod immune genes that normally limit infections by Gram-negative bacteria. However, due to the additional invasion of the salivary glands of the body louse and its transmission through saliva in case of high infectious doses (>10^7^ colony-forming units/mL) an important role in plague epidemiology has been postulated [41,233,237].

### 5.2. Effects of the Vector on the Bacteria—Development of Bacteria in the Fleas

#### 5.2.1. Development of *Rickettsia* and *Bartonella* in Fleas

*Rickettsia* sp. ingested during feeding attach themselves to the epithelial cells of the midgut and infect them [8]. However, *Rickettsia* sp. are not restricted to the flea midgut. Rather, they infect epithelial cells of the midgut and then disseminate throughout the flea body via the hemocoel [4]. Tick-borne rickettsiae (typhus group) use so-called rickettsiae-induced actin-mediated motility, which is based on pathogen-regulated actin polymerization and rearrangement, to move within cells and infect neighboring cells [9,92]. The typhus group, including *R. typhi*, does not contain the genes required for this manipulation of the host cell and is released from the cells by cell lysis [8,9]. It then infects other tissues, including salivary glands, ovaries, hindgut and fat body [8]. However, *R. typhi* remains primarily in the digestive tract during flea infection and causes lysis of epithelial cells, releasing bacterial cells back into the midgut lumen, which are subsequently excreted in the feces [8,64]. Viable *R. felis* are also found in flea feces [58]. *Rickettsia* sp. can remain infectious in excreted flea feces for several months [8].

The interactions of *Bartonella* species with fleas are largely unknown, but *B. henselae* is a strong biofilm former. The adhesion BadA, which is homologous to *Yersinia* adhesinA, appears to induce autoaggregation and may play a role in immune evasion in mammals, but may also be involved in biofilm induction [66,94]. This may also promote persistence in fleas [221]. The development of *B. henselae* is restricted to the intestinal tract and the bacterium is excreted in the feces, but other species also invade the hemocoel [20,21].

#### 5.2.2. Development of *Yersinia pestis* in Fleas: General Aspects

Vectors have evolved various strategies to combat pathogens ingested with blood. Firstly, the conditions in the midgut, including pH, osmolarity, lysed red blood cells, digestive enzymes, harmful blood digestion by-products and the antibacterial response, pose a challenge/stress to some microorganisms [61,62,115,178]. The flea intestine in particular is considered an unfavorable environment for Gram-negative bacteria. Excretion by the vector poses a particular risk to *Y. pestis* [62]. *Y. pestis* appears to be protected against toxic components present/produced in the flea gut, including yersinia murine toxin and ribose-5-phosphate isomerase [62,82,112]. Some of these factors are also relevant after flea ingestion of *Y. pestis* during blood feeding on infected hosts. The pH and osmolarity of the midgut of fleas only change within 24 h after blood ingestion and then recover [116] (see Section 3.4). Apart from iron, which is not limited in the flea [114], there may be a zinc limitation in the flea vector, which appears to be overcome by yersiniabactin. The ability to bind zinc is not a prerequisite for colonization of the flea midgut [238,239], but a yersiniabactin mutant strain that cannot bind zinc ions does not colonize the flea at the same rate as the wild type [119,120]. During blood digestion, the copper concentration also increases in the flea gut. This is probably sensed by a two-component system. The activated system helps *Y. pestis* to improve flea gut colonization [117].

In the flea, *Y. pestis* adapts its metabolism to switch from sugar as the carbon source and most likely uses peptides and amino acids as its primary energy source [61,77,240]. Furthermore, the vector-host-vector transition/cycle leads to sudden enormous temperature changes. In *Y. pestis*, several hundred genes are differentially regulated at 26 °C and 37 °C [77,121]. In vitro, more genes (approximately 60%) are upregulated at 26 °C than at 37 °C (approximately 40%) [121], which may indicate that adaptation to the flea vector is complex. In the *Trypanosoma* sp. parasite-vector system in tsetse flies, the drop in temperature induces rapid reprogramming of gene expression, allowing the parasite to develop into forms that can successfully infect the midgut. Furthermore, this parasite is known to be preadapted for survival in the insect vector. This is based on a quorum sensing system and a response to a high parasite density in the host blood [153].

Quorum sensing, a form of density-dependent cell-to-cell communication, also enables bacterial populations to synchronize gene expression [241]. *Y. pestis* is also capable of quorum sensing [242]. Some evolutionary adaptations such as the expression of the F1 capsule in the mammalian host facilitate flea infection by creating a high bacteremia [122], but it is unknown whether a form of cellular preadaptation occurs in the mammalian host that promotes subsequent colonization by the flea. In this context, bacterial genetics and metabolism are simpler compared to protozoan parasites, as exemplified by the immediate response of the *Y. pestis* thermosensory system upon uptake by the vector to trigger the necessary adaptations [34]. Considering the other side of the vector-host interface, passage through the flea vector is thought to prepare *Y. pestis* for mammalian innate immunity [240]. A potential preadaptation could involve gene expression leading to products that facilitate the infection process in mammals. This could include an adhesin, which is expressed four times higher in *Y. pestis* than in *Y. pseudotuberculosis* [239]. This adhesin is an important virulence factor in the mammalian host. It confers serum resistance by binding to and blocking components of the complement system [124]. The expression of a factor that is not required at this stage in the flea could be a preadaptation to withstand the early innate immune response in the mammalian host [239]. In addition, extracellular components of the biofilm matrix are transferred together with *Y. pestis* into the bite site and can support the course of infection compared to injection with a needle [62,81]. Since the biofilm consists of a large number of host-derived compounds, it could also be a strategy to disguise the initial immune response [243].

#### 5.2.3. The Course of the Development of *Yersinia pestis* in Fleas: Importance of Bacterial Factors

A high bacterial load, up to 10^9^ colony-forming units per ml which can develop in some rodent species, generally favors successful flea infection [33,77,81]. Depending on the flea species, an average of at least 10^6^ to 10^7^ cells per ml is required for successful flea infection, although individual fleas may still be able to clear such a high or even higher infectious doses [51,62,150,196,244]. In infected humans, the number of viable bacteria in the blood varies widely, amounting to several orders of magnitude [245]. Considerable variability is also observed after experimental infections between different rodent species, with pathogen concentrations in individual rats both below and above the aforementioned threshold allowing transmission [246]. The obligatory association between *Y. pestis* and fleas led to positive selection in hosts with high levels of bacteremia [32,35,62,150,151]. This is an evolutionary trend, although a single point mutation resulting in an amino acid substitution in the plasminogen activator found in all recent *Y. pestis* isolates resulted in reduced cleavage of the yersinia murine toxin and thereby limited the lethality of *Y. pestis* to hosts [247], suggesting a balanced evolutionary fine-tuning of *Y. pestis* virulence and lethality.

After infection, *Y. pestis* multiplies in the midgut and significantly in the foregut [62,77]. Midgut growth may not be necessary for transmission and the proventriculus is now considered the primary site of infection with biomass formation within one hour p.i. [61,82,86]. In contrast to this established understanding of cyclic horizontal vector-host transmission, a recent study suggests that *Y. pestis* spreads into the reproductive tissue of both male and female fleas, which is detectable several months after ingestion of infectious blood. *Y. pestis* appears to be located in testes and ovaries, and the number of viable cells obtained from body tissues of successive life cycle stages increased from larva to newly emerging offspring [50]. This new concept of vertical transmission—similar to the transmission of *R. felis* by fleas [58] (see Section 4.2.1)—could be groundbreaking in explaining missing parts of the current understanding of plague epidemiology and ecology, but definitely requires further confirmation, also in other flea species and strains of *Y. pestis*. This new mechanism, in addition to transovarial (vertical) transmission, may have a second major advantage with regard to the long-term survival of *Y. pestis*. If uninfected larvae feed on *Y. pestis*-infected eggs, this could lead to additional horizontal transmission within the flea population during the larval stage. However, this requires further investigations. First, it must be demonstrated that uninfected larvae become infected when feeding on *Y. pestis*-infected eggs, and if so, where the infection is localized and whether this can lead to adult fleas capable of transmitting *Y. pestis* through the primary transmission route of blockage and regurgitation.

Many of the molecules involved in *Y. pestis*-vector interaction have been identified and characterized in recent decades [61,62]. One of the key molecules for successful colonization of the midgut is the plasmid-encoded yersinia murine toxin [112] (see Section 2). Mutants lacking this phospholipase D activity show signs of impaired membrane integrity and are rapidly eliminated from the midgut [77,112]. The importance of this toxin for profound flea infections depends on host blood. For example, it is required to ensure flea infection and transmission when fleas are infected with blood from black rats or humans. Conversely, flea infection rates of yersinia murine toxin mutants in the blood of brown rats are similar to those of the wild type [85]. It also promotes early aggregation of *Y. pestis* cells in the foregut of fleas [82,85,243]. Insertion of this yersinia murine toxin gene into *Y. pseudotuberculosis*, which is otherwise found in the hindgut during experimental infections, allows colonization of the midgut with a bacterial load comparable to that of *Y. pestis*, but results in neither blockage nor strong transmission. However, *Y. pseudotuberculosis* mutants containing all the mutations that allow efficient transmission of *Y. pestis* by fleas are transmitted similarly to the wild type of *Y. pestis* [104].

Finally, *Y. pestis* is defecated [61]. A clearance of *Y. pestis* by individual fleas despite high infection inocula > 10^8^ cells/mL depends on the blood source, with higher numbers of clearance in *Oropsylla montana*, a flea of the ground squirrel, when infected by and subsequently fed on mouse blood compared to rat blood [51]. Another consideration is the feeding and defecation rate. Frequent-feeding flea species (e.g., *C. felis* and *X. skrjabini*), which start excreting soon after ingesting blood, can clear the infection more quickly within the first few days than other flea species, which feed less frequently and digest the blood more slowly [61].

Summarizing the development of *Y. pestis* in the fleas, the bacterium requires a high bacterial load in the ingested blood for a successful infection. It then colonizes all regions of the intestinal tract, with the proventriculus being the most important region for transmission.

## 6. Transmission of *Yersinia pestis* by Fleas

### 6.1. Different Modes of Transmission of Yersinia pestis

The main vectors of *Y. pestis* are fleas. Other insects can also transmit the bacterium, but to a lesser extent, e.g., triatomines and lice [233,248]. *Y. pestis* infects mammals mainly via backflow/regurgitation during flea bites. Saliva components are unlikely to favor early *Yersinia* infection in mammals, as no effect on transmission or pathogenesis has been observed in mice [249], but saliva pumped during blood ingestion may carry away *Y. pestis* cells that have resided in the salivary grooves since the last ingestion of infectious blood [250]. Another possible mode of transmission is mechanical transmission, but *Y. pestis* remains viable on mouthparts for only a few hours [41]. In addition, fleas can rapidly digest and defecate viable *Y. pestis* cells when ingesting blood, which can then infect new human hosts, e.g., via skin lesions [41,250].

More important are two transmission modes based on foregut obstruction thereby causing backflow or regurgitation of *Y. pestis* from the alimentary tract during blood feeding, early-phase transmission and transmission by foregut blockage [35,59,61,62,80,232]. The traditional transmission model by blocked fleas was described over a century ago [251]. In contrast, early-phase transmission was only accepted as an alternative mode of transmission much later [35,232], and has been shown to be biofilm-independent, but the exact mechanisms remained unclear [232,250].

In the past years, a new model has been proposed in which the physiological basis for both transmission modes (early phase and blockage) is the same, with the difference that the early accumulation of biomass in the foregut is not strong enough to withstand the shear forces of the blood that is ejected during later blood uptake and is thereby transported to the midgut, whereas the biomass rebuilt in later stages of the infection process is firmly embedded by biofilm formation and resists the mechanical shear forces during subsequent blood uptake. Thus, a smooth transition between an early stage with cell aggregation (in which early-phase transmission may occur), via a later biofilm-mediated partial blockage and a late stage with a complete blockage is emphasized [61,62,80,82,85,243].

The strategy of narrowing or blocking the foregut lumen to facilitate its own transmission is not restricted to *Y. pestis*, as the flagellate *Leishmania* sp. uses a similar strategy in sandflies [252]. In this system, attachment to the midgut epithelia is initially important to induce the necessary developmental changes to complete the life cycle [153]. While *Yersinia* tends to impair the physiological function of the foregut through clogging, *Leishmania* spp. use similar mechanisms, but also enzymatically damage the cuticle of the cardia and open it to the esophagus, disrupting the blood supply and promoting backflow or regurgitation [253].

### 6.2. Formation of Biofilms by Yersinia pestis

Biofilm formation by *Y. pestis* is an important aspect of transmission. Bacterial biofilms are a mash of extracellular polymers secreted by the bacteria, containing proteins and extracellular DNA, in which colonies are embedded [254]. This helps the bacteria to survive, especially in nutrient-poor and hostile environments. The classic concept of biofilm formation includes several stages, beginning with the attachment of planktonic cells to a surface, the irreversible attachment and maturation of the biofilm, including cell growth and the production of extracellular polysaccharides, and finally the dispersal of parts of the biofilm. Depending on the environment, however, this may be an oversimplification, and there are other mechanisms as well [255]. In contrast to the classical concept, early aggregation of *Y. pestis* in the flea vector occurs predominantly intercellularly without profound attachment to the gut surface [61,62,82,239]. However, a certain degree of adhesion can occur, e.g., due to hydrophobic interactions [61,83]. According to recent studies on the composition of *Y. pestis* biofilms in vivo, the cells do not adhere to each other even in early stages but become entangled in the viscous matrix [243]. In comparison, non-surface-adherent aggregates of *Pseudomonas aeruginosa* are known that exhibit similar growth kinetics and resistance to surface-adherent biofilms [256].

The initial tendency to aggregate can have various causes. For example, cell aggregation in *P. aeruginosa* can be induced by either polysaccharides or extracellular DNA, whereby polysaccharide-producing cells do not aggregate with non-producing cells [257]. In this bacterium, too, polysaccharides and extracellular DNA interact to form extracellular fibrous structures, which are fundamental components of biofilm structure [258]. The tendency of individual *Y. pestis* cells to aggregate is also detectable in vitro in some culture media [67]. However, the aggregation mechanism is not the same as in in vivo [239]. Aggregation in the flea gut is probably triggered spontaneously by the phenomenon of depletion aggregation [243]. This results in viscous brown proventricular masses in which *Y. pestis* cells are embedded very soon p.i. in the proventriculus and then increasingly in the midgut [61,82,243].

In stationary, surface-attached bacterial biofilms of *P. aeruginosa*, the cell population is not a single unit with homogeneous properties. Rather, it consists of subpopulations, arrested cells that thicken the biofilm, and motile cells that expand the biofilm area [259]. The specialization of *Y. pestis* cells in flea biofilms is unknown; however, the cells are unevenly distributed within the aggregates in the proventriculus. Furthermore, extracellular DNA does not appear to play a role in *Y. pestis* biofilms in fleas [243].

In some bacteria, such as *Pseudomonas* species, the formation of a strong biofilm depends on two quorum-sensing systems. Without these genes, biofilm formation is only moderate [260]. These systems initiate the production of cyclic diguanylate monophosphate (di-GMP) in response to surface attachment, which triggers the secretion of extracellular polymeric substances [259,261]. However, quorum sensing does not generally induce biofilm formation in bacteria but rather coordinates maturation and decomposition of the biofilm [241].

In *Y. pestis*, the hemin storage (hms) genes are responsible for biofilm formation in the proventriculus, resulting in a transmissible infection. They are similar to the genes of other biofilm-forming bacteria. Hms-negative *Y. pestis* infect the midgut but do not colonize the proventriculus and are unable to form biofilms [59,76,77,82,238]. However, the effect is not absolute, as a specific *Y. pestis* strain deficient in the hms gene is also detected in the proventriculus of fleas, albeit in lower abundance and spatial extent than wild-type strains. This suggests that the hms operon overall increases the proportion of fleas capable of transmitting *Y. pestis* and improves long-term persistence within the flea population [86]. *Y. pestis* can also cause biofilm-mediated blockage in the cat flea; however, other regulators appear to be involved in biofilm regulation, and the mechanism is still unclear [262].

The ability to form biofilms in fleas requires specific properties. The regulatory mechanism appears to have evolved in a complex and unique way in *Y. pestis*, intertwined with and dependent on other genetic changes that accumulated after the separation from the ancestor, including gene losses and a reduction in the cyclic di-GMP regulatory system [62,263]. For example, despite being potent biofilm formers, *Y. pseudotuberculosis* strains do not produce biofilms in the rat flea *X. cheopis* in vitro and do not colonize the foregut [264]. In turn, a genetically modified *Y. pseudotuberculosis* mutant capable of producing biofilms and blocking the foregut of fleas does not increase expression of the hms gene, but, unlike *Y. pestis*, strongly upregulates another secretion system for secreting biofilm components in the flea. Consequently, this mutant loses the ability to block the gut when this secretion system locus is deleted [239]. *Y. pestis* also possesses these secretion system loci, but the genes are downregulated during flea passage [239,240].

As in *Pseudomonas*, extracellular matrix production in *Y. pestis* is controlled by the concentration of cyclic di-GMP via the hms gene products [59,62,76]. In bacteria, the cyclic di-GMP system is complex and can, among other things, sense light, redox potential, temperature and abiotic chemicals [265]. For example, a guanylate cyclase that controls cyclic di-GMP in *P. aeruginosa* regulates the maintenance of biofilm formation in the presence of hydrogen peroxide [266]. The intracellular cyclic di-GMP of *Y. pestis* is synthesized by two guanylate cyclases encoded by the hmsT and hmsD genes and degraded by a phophodiesterase encoded by the hmsP gene [263,267,268], but these molecules are subject to finely tuned regulation, including a three-component hms control system, which is important for flea blockage [269,270,271]. However, it is also expressed at elevated temperature but regulated post-transcriptionally [239]. Part of the molecular regulation is a specific chaperone, which represses hmsT synthesis post-transcriptionally and activates the cyclic di-GMP degrading phosphodiesterase [272]. In contrast to these results, mutants deficient of this chaperone are unable to block the foregut in vivo, indicating that this chaperone may positively support biofilm formation [273]. However, the response may be dependent on specific environmental or nutritional conditions [272]. A specific RNA-binding protein, in turn, regulates the repression of hmsT mRNA translation by the specific chaperone, thereby stimulating high cyclic di-GMP levels. This RNA-binding protein is also involved in carbon metabolism and could influence the fundamental nutritional transition in the flea midgut by triggering biofilm formation [274]. In the flea gut, hmsD is considered the major enzyme, while hmsT activity is of minor importance [263,267]. There, a positive regulator, hmsE, activates hmsD [270]. This activation is based on the specific RNA binding protein, which binds to the hmsE mRNA and stimulates translation, which in turn promotes hmsD-mediated biofilm formation [271]. Consequently, *Y. pestis* lacking the specific RNA binding protein exhibit reduced flea blockage and transmission capacity [274]. Another component of the hms complex involved in biofilm formation is hmsB [275].

There are various genes and molecules involved in biofilm induction and regulation, but their exact contribution and interplay are not fully understood [61,62]. The regulatory network includes a transcriptional regulator that stimulates the expression of a specific operon and the three-component hms system [276]. Another positive regulator is a small regulatory RNA which also activates, for example, the specific hms operon and the three-component hms system thereby stimulating biofilm synthesis [277]. In *Y. pestis*, environmental cues are also promising stimuli/inducers, as passage to the vector is accompanied by a fundamental change in the microenvironment. Two of these factors, pH and osmolarity, fluctuate in the flea gut at different times but do not stimulate the expression of specific genes of a two-component system. This system appears to detect nutrient scarcity during ongoing hemolysis and proteolytic digestion which may induce biofilm formation [278]. Another two-component system, PhoP-PhoQ, may sense the lower pH in the flea gut [279]. The PhoP-PhoQ plays an important role in both the mammalian host and the flea gut [115]. The transcriptional regulator of this signal transduction system is PhoP [240]. Although PhoP does not have a direct regulatory effect on the hms genes [280], it promotes diverse physiological adaptations including changes in the outer cell surface and PhoP-negative mutants can infect fleas with normal bacterial loads but form less cohesive biofilms resulting in a great reduction in foregut blockage [114,115,279].

A general regulator is highly expressed in the flea gut [240] and presumably also senses environmental signals in the flea midgut, as mutants lacking the general regulator gene can infect fleas and produce biofilms and transmission, but are inferior in competition with wild-type strains during coinfections. However, when this regulator is overexpressed, the mutant outperformed the wild type in a nutrient-dependent manner [281]. This regulator activates virulence genes in *Y. pseudotuberculosis* [282], where it regulates the transition from planktonic bacterial growth to biofilm formation in response to dietary changes [283]. In *Y. pestis*, this general regulator activates, among others, the specific hms operon as well as the genes of the three- component hms system and downregulates hmsP, thereby stimulating biofilm production in a temperature-dependent manner, with maximum expression of the general regulator in vitro at 26 °C compared to 37 °C [280]. This suggests a key role in coping with the drop in temperature following ingestion by fleas [77]. Specifically, hmsB is activated by a LysR-type regulator, but the exact molecular mechanism is still unknown [272].

Furthermore, biofilm formation by *Y. pestis* in the flea is enhanced in reducing environments [284]. The underlying molecular mechanism presumably leads to reduced concentrations of hmsC, a negative regulator, and reduced binding of hmsD, which stimulates biofilm formation [285]. A small RNA may also be involved in sensing environmental stimuli to regulate biofilm production [272].

In summary, biofilm formation by *Y. pestis* in fleas is a common bacterial phenomenon and is regulated by hms genes. In *Y. pestis*-infected fleas, differences from the mammalian environment, in particular a temperature shift and environmental cues such as nutrient deprivation during ongoing digestion, are sensed by a limited number of signal transduction systems and either induce the expression of these genes or support normal biofilm formation.

### 6.3. Early-Phase Transmission of Yersinia pestis

Early-phase transmission occurs rapidly within a few hours to a week [35,196,232,250,286,287,288,289]. It is most effective immediately after ingestion of infectious blood and then declines over the first days [286,287,290,291], indicating that it is transient and not self-reinforcing, as would be expected if it is caused by stable, localized, persistent bacterial growth. This mode of transmission is most pronounced during the first feeding after infection [35,61,287,291]. Thus, the time window for early-phase transmission within the first week is clearly influenced by feeding frequency.

Regarding temperature, early-phase transmission does not occur at 10 °C and these transmission rates of *X. cheopis* are not significantly affected at temperatures between 23 °C and 30 °C [289]. Transmission is based on bacteria that remain in the foregut/proventriculus, aggregate and interfere with blood uptake. The cells form aggregates as a dark brown mass on and between the proventricular spines [61,62,80,82,86,239,262].

The brownish thick biomass in the proventriculus can be observed as early as one hour after ingestion of *Y. pestis*-containing blood, but not in case of uninfected blood [61,82]. The aggregation is most likely triggered spontaneously by depletion aggregation which occurs in environments with high concentrations of, e.g., biomolecules or bacteria [243]. The aggregates consist of *Y. pestis* and blood and vector components. When infected with mouse blood, the protein content of these aggregates 24 h p.i. consists of about 72% mouse proteins, most of which originate from hemoglobin and blood cell membranes (around 64%), 20% *Y. pestis* proteins, and 8% flea proteins. In addition, the aggregates in fleas infected with mouse serum contain a particularly high concentration of cholesterol [243].

Early-phase transmission is usually observed in naturally infected flea species [132]. For example, *O. montana* can transmit *Y. pestis* by early-phase transmission [35,51,196]. However, there are also marked differences in the efficiency of different species of the genus *Oropyslla* or other flea genera [287]. In *X. cheopis*, about half of the proventriculus is colonized two days p.i., which increases to about 90% one and two weeks p.i. [86]. In other studies, the intensity of proventricular colonization peaks at day 3 days p.i. Furthermore, colonization depends on the blood source, with pig and rat blood resulting in more frequent and intensive colonization compared to mouse and prairie dog blood [50]. The anterior part of the proventriculus is the intersection to the esophagus and is colonized early on by bacterial aggregates. This appears to be the main site where proventricular blockage is initiated. The stabilization of the biomass and complete colonization of the proventriculus reinforce the blockage [82,86].

The efficiency of early-phase transmission depends on the experimental design, the host or blood source, but also on the characteristics of the vector. Regarding experimental design, early-phase transmission is also called mass transmission since most transmission studies used numerous (>5 up to over a dozen) infected fleas per individual to determine the transmission efficiency [51,61,128,196,292]. This may reflect natural infestation rates and is necessary as experiments using single fleas yield only a poor transmission success around 2% [35] but may also overestimate transmission rates.

Regarding blood source, infection of fleas with rat blood results in higher transmission efficiency than infection with mouse blood [80]. For *O. montana*, approximately 3% of fleas infected with mouse blood can transmit *Y. pestis*, compared to approximately 10% when infected with rat blood [51]. Furthermore, certain blood sources, such as blood from brown rats or guinea pigs, when they contain *Y. pestis*, stimulate a physiological response in some fleas (*X. cheopis* and *O. montana*), the so-called post-infection esophageal reflux, which is more severe than with other blood sources (e.g., mouse blood), disturbing the blood feeding and causing backflow of blood components into the esophagus, thus facilitating transmission [80]. The effect correlates with larger *Y. pestis* aggregates in the proventriculus, which are associated with blood components such as erythrocyte stroma and hemoglobin crystals and can spread to the esophagus. Furthermore, post-infection esophageal reflux is associated with the rate of blood digestion, with slow digestion and strong crystal formation favoring this effect. In some fleas, such as *X. cheopis* and *O. montana*, this reflux induces higher shedding of bacteria during the early-phase transmission and, in addition, rapid biofilm-mediated transmission within a few days. However, this effect does not occur in all flea species, as cat fleas infected with rat blood do not exhibit post-infection esophageal reflux [51,80,85,128]. *C. felis* is known to have only poor vector competence but may be capable to transmit *Y. pestis* through early-phase transmission [230]. Contradictory to the current model, the observation that *C. felis* could transmit *Y. pestis* within the first week by early-phase transmission despite daily feeding [128], is not fully conclusive with the continuous removal of insufficiently stabilized *Y. pestis*-biomasses by incoming blood during subsequent ingestions of blood. However, the number of colony-forming units (CFU) that have been expelled is low [128]. It should also be noted that assessing transmission efficiency based on CFU egested during feeding and counted using plating methods does not necessarily measure true transmission efficiency. For example, in mouse infection experiments, bacteria mostly remain in the skin or lymphatic system, and terminal bacteremia rarely develop in flea-infected mice [81].

While cells adhere to each other through aggregation and extraneous substances, the transmission does not initially rely on the hms gene-induced formation of a biofilm [239]. For example, biofilm-deficient *Y. pestis*-strains are efficiently transmitted by early-phase transmission [293] and the yersinia murine toxin, which is necessary for both, midgut colonization and late-stage blockage, is not required for early-phase transmission when rat blood is used for infection [126]. Nevertheless, it also plays a role in early-phase transmission as it enhances the early aggregation of *Y. pestis* cells [243].

The early-phase transmission depends on the initial infectious dose ingested by the flea and usually does not occur below a threshold value of 10^8^ CFU/mL [294]. Interestingly, early-phase transmission does not correlate with the pathogen load from fleas, which is a decisive factor for later transmission through the “blocked flea mechanism” [35,126,293]. However, this assessment is based on CFU per flea determinations which may not be the relevant parameter for a correlation as the ratio of the CFU load between proventriculus and midgut in these studies remains unknown. In addition, when the minimal infectious dose is assessed by a suitable study design, at least a clear trend towards higher early-phase transmission rates after infection with higher infectious doses is detectable [294]. Non-attached and loose *Y. pestis*-aggregates tend to be mechanically torn off by the incoming blood and transported to the midgut, thus terminating early-phase transmission [82].

### 6.4. Foregut Blockage Transmission of Yersinia pestis

The classical mode of transmission is based on increasing blockage of the foregut lumen, aggravated by the formation of biofilms [61,238]. It is observed at the earliest two days after ingestion of infectious blood (and can therefore overlap with early-phase transmission), frequently after four to five days and most pronounced after one to three weeks [51,80,126,196,244,250]. Initially, this form of transmission was observed only in a few naturally infected flea species [35,132]. However, it is the main mechanism in infected *X. cheopis*, the most important plague vector worldwide. In addition, other flea species that were previously considered poor blockers were also blocked using new study designs [51,196,228]. After ingestion with the blood, the bacterium multiplies in the midgut and can reach 10^6^ CFU per infected flea after one week but does not attach firmly to the proventriculus or the midgut epithelium [61,62,77,239]. Large biomasses also develop in the foregut, narrowing and obstructing the lumen [62,82]. Colonization is most pronounced in the anterior part of the proventriculus, but bacterial load may not be the primary factor causing blockage [86]. Furthermore, the anterior colonization in *X. cheopis* is not limited to the proventriculus and can extend from there to the esophagus and mouthparts in some *Y. pestis*-strains [50,84,228]

The required infectious dose also depends on host factors. For example, *X. skrjabini* requires a higher infectious dose than *X. cheopis*, but blocks earlier (three days p.i.), albeit the percentage blockage rate is lower [295]. In addition, the infection rates are also influenced by the blood source [80]. The blood source has a variable impact on different transmission modes of the same infected flea. For example, mouse blood strongly reduces early-phase transmission in *O. montana* (see Section 6.3) compared to rat blood but has a less pronounced effect on transmission by blockage [51].

Different strains or biovars of *Y. pestis* differ in their mode of foregut colonization and their dominant location. However, these differences in the strains do not affect the transmission efficiency of the rat flea *X. cheopis*. Despite the observed strain-specific localization and obstruction of the foregut [84], the molecular pathomechanisms remain unknown. Foregut colonization appears to be facilitated by lipoylation of host enzymes, involving a *Y. pestis* enzyme that catalyzes steps of lipoate biosynthesis [296]. Furthermore, yersinia murine toxin, beyond its protective function during flea infection [61,112] appears to be important for early aggregation, which also contributes to foregut colonization [243]. Ribose-5-phosphate isomerase appears to be involved in early biomass stabilization, which is later consolidated by biofilm formation, as double mutants do not block fleas [82]. After *Y. pestis* reaches up to 10^6^ CFU in infected fleas after one week, the bacterial load consolidates and remains stable [51,62,77]. This is sufficient to maintain the flea-host cycle, as fleas with such a load become efficient vectors due to classic flea blockage [297]. In an advanced stage of flea infection, *Y. pestis* induces biofilm formation in the foregut, which impairs blood uptake and can also cause blockage there. As a result, fleas can no longer suck blood efficiently and tend to regurgitate intestinal contents along with *Y. pestis* into the host wound [59,77].

Pathogen transmission is already increased in partially blocked fleas, some of which subsequently develop a complete blockage [61]. The development of blockage also depends on feeding behavior and is not an absolute characteristic of a flea species. Some fleas, such as the cat flea *C. felis*, frequently ingest blood [140], while other flea species feed less frequently [128]. This behavior influences the formation of *Y. pestis*-induced biofilms. Growing bacterial aggregates are initially susceptible to removal by ingested blood due to the lack of irreversible attachment to the proventricular cuticle. These results are also consistent with the new model of a smooth transition between early-phase transmission and blockage, at least in part, as stable colonization of the cat flea *C. felis* and transmission by blockage (but initially by early phase transmission) do not occur when the flea is fed daily after infection. Therefore, with frequent blood feeding, the premature biofilm is more likely to be repeatedly translocated to the midgut, and a lower percentage of cat fleas retain an infected proventriculus. Twice-weekly feeding the cat flea increases the blockage rate, most likely because the aggregates have more time to be stabilized by biofilms. A typical effect of frequent daily feeding is also that a higher percentage clears the *Y. pestis* infection through rapid digestion and defecation [61,82,128]. In the rat flea *X. cheopsis*, a less frequent feeder, the biofilm induced by *Y. pestis* has more time to mature and stabilize, increasing the likelihood of withstanding the shear forces during subsequent blood uptake [82,128].

While *X. cheopis* shows strong colonization of the proventriculus shortly after ingesting the blood, and the blockage and transmission rates are the highest among fleas, *O. montana* can also efficiently block, but usually shows weaker colonization of the proventriculus in the first few days [77,126,196,244]. *Oropsylla hirsuta*, another vector associated with wild rodents in the USA, can also transmit *Y. pestis* by blocking the foregut within the first two weeks with similar efficacy to the early-phase transmission, although the blockage rate of approximately 10% and the transmission efficiency are not as high as those of *X. cheopis* and *O. montana* [228]. In contrast, *P. irritans* rarely becomes blocked [36]. However, this also depends on the blood source. Transmission efficiency increases when *P. irritans* feeds on experimentally infected rat blood compared to infected human blood, which causes foregut blockage significantly less frequently [49].

Within an infected population, the flea blockage rate may be low at any given time, for example, approximately 2% of fleas on prairie dogs after an epizootic [297]. According to model calculations, however, the blockage rate at the beginning of an epizootic can be significantly higher. Further studies with infected field populations are needed to determine dynamic changes in the blockage rate during an epizootic [132].

### 6.5. Comparing Efficiency of Transmission Modes

Comparing both transmission routes, early-phase transmission is generally considered less effective than blockage [51,61,62]. Both *X. cheopis* and *O. montana* can efficiently transmit *Y. pestis* through foregut blockage with high CFU counts, but only poorly through early-phase transmission [196]. Some studies demonstrate the superiority of early-phase transmission with the prairie dog flea *O. hirsuta* [291], while others show effective transmission through blockage under similar conditions [228]. The blockage mechanism is also more efficient in *O. montana* when compared within the same study under standardized conditions and may have been underestimated in previous studies due to increased stress-related flea mortality [51,196].

Furthermore, early-phase transmission requires a high infectious dose (≥ 10^8^ CFU/mL) [292], which may limit its role in less susceptible mammalian populations that typically do not reach these bacteremic levels [61]. In general, the estimated transmission rate for early-phase transmission ranges from 0 to 10% among different flea species, with only a few CFUs excreted [61,196]. In comparison, a single feeding attempt of a blocked flea has a transmission success of up to 50% [150,196]. Regarding experimental design, higher transmission capacity is observed after flea feeding on live hosts than in artificial systems [295]. The results may also be affected by the selection of strains and vector species, rearing and adaptation in the laboratory, sampling times, and many other variables. Although the percentage of fleas that are potent vectors by early-phase transmission is clearly lower compared to partially or blocked flea transmission in *O. montana* when fed on rat blood, the absolute median number of CFUs injected by individual fleas is similar between the different transmission modes [51].

Despite the general superiority of blockage, early-phase transmission has been suggested to initiate and maintain epizootics among animals [35,232,291]. However, there is some controversy regarding the role of early-stage transmission in epizootic diseases: According to modelling, it may also lead to increased resistance in some individual hosts through the transmission of sublethal doses, potentially limiting epizootic diseases and supporting the transition to enzootic circulation [51,81].

In summary, both transmission modes of *Y. pestis* are important. Early-phase transmission is possible within a short time after ingestion of infectious blood. More important is the progressive obstruction of the foregut by a mass of bacteria and blood components embedded in a biofilm. When the flea attempts to ingest blood, the bacteria from this mass enter the mammalian host.

## 7. Effects of the Bacteria on the Fleas

### 7.1. Effects of Bacteria on the Immune Response of the Fleas

In the absence of pathogens, blood feeding alone induces transcriptional changes. Several factors such as serine proteases and defensins are upregulated by fleas [161]. The cellular immune response of fleas can only act against bacteria that invade the hemocoel, e.g., *Rickettsia* and *Bartonella*, and not against *Y. pestis*, which remain mainly confined to the intestinal lumen of the flea. In the cat flea, several transcripts, including trypsin- and chymotrypsin-like factors, are differentially expressed by *R. typhi* infections compared to ingestion of non-infectious blood [8]. *Rickettsia* sp. are recognized and combated by the Imd pathway, and knockdown of important transcripts and the Imd pathway leads to increased *R. typhi* load in the cat flea [174]. This response is restricted to some AMPs, and defensins are not upregulated in response to *R. typhi* [174,298]. Furthermore, the presence of *R. felis* in the salivary glands induces an upregulation of the expression of immune genes, including AMPs such as defensin-2 [55]. However, the flea response to rickettsial infection is still poorly understood and difficult to study experimentally [8,298]. Surprisingly, infection of the cat flea with *B. henselae* does not lead to upregulation of transcripts of key components of the immune signaling system, including the Imd pathway, up to 24 h after infection. However, antimicrobial activity in the midgut is increased, indicating that this pathogen may induce other signaling pathways. In contrast, other bacteria such as the Gram-positive *Micrococcus luteus* and the Gram-negative *Serratia marcescens* induce a strong measurable transcriptional response of the Imd pathway [129]. The genus *Serratia* belongs to the family Yersiniaceae and is taxonomically relatively close to *Y. pestis* [69].

The presence of *Y. pestis* in ingested blood is sensed by the flea, as evidenced by changes in gene expression in *Xenopsylla*, with up to several dozen transcripts being differentially expressed when fleas are fed *Y. pestis*-containing or non-infectious blood [83,299]. In particular, two peptidoglycan recognition proteins, which are strong activators of the Imd signaling pathway, are upregulated in response to *Y. pestis* in the blood [83]. The upregulation of these genes after blood ingestion [161] could also modulate the flea’s immune response to the presence of other bacteria (see Section 3.5). This induction of the Imd signaling pathway leads to the synthesis of humoral effectors directed against these Gram-negative bacteria [83,171,300]. The rat flea *X. cheopis* responds to the presence of a high load of *Y. pestis* (>10^8^/mL) with >30 transcripts that are differentially regulated compared to unfed fleas or fleas fed with blood that does not contain *Y. pestis*. These include immune genes associated with the Imd pathway involved in AMP production [83]. The rat flea produces attacins that act against *Y. pestis* and inhibit colony formation in vitro. After upregulation by the presence of *Y. pestis* in the blood, the bacterium continues to multiply, but then lysis takes place [179]. At low temperatures in the flea midgut, *Y. pestis* is resistant to cationic AMPs mediated by the PhoP-PhoQ system, which is upregulated in the flea and modifies surface components including lipopolysaccharides [77,240,301]. Infection with an AMP-sensitive *Y. pestis* mutant results in a higher percentage of fleas killing *Y. pestis* seven days p.i. and a lower bacterial load compared to the wild type [127].

The pathogen resists the flea’s innate immunity through the PhoP-PhoQ system in which the regulator PhoP is activated by various stress factors sensed by an inner membrane sensor, PhoQ. This kinase then regulates global gene expression [115]. In addition, mutants negative for the regulator or carrying a single amino acid substitution in it, although able to colonize the flea midgut, show reduced biofilm formation and blockage of flea vectors, negatively impacting transmission efficiency or at least causing fitness defects when coinfected with the wild type [114,302]. *Y. pestis* has also evolved other mechanisms to evade the destructive effect of some AMPs. For example, an intrinsic enzymatic activity that leads to a biochemical modification of the lipopolysaccharide coat of the outer membrane protects *Y. pestis* from ceopin, an AMP from *X. cheopis*. While this antimicrobial peptide shows strong activity against Gram-negative bacteria and *Y. pestis* strains without arabinose modification, the antimicrobial peptide cannot bind to and act efficiently against wild-type strains [301,303].

In addition to inducing the Imd signaling pathway, which leads to the production of AMPs against Gram-negative bacteria [83], *Y. pestis* infections increase the concentration of reactive oxygen species in the midgut, and bacterial load increases after feeding of antioxidants, likely through the scavenging of reactive oxygen species [299]. In infected *Xenopsylla*, only a few transcripts are upregulated, presumably including genes for reactive oxygen species [299]. Similar mechanisms are evident in the midgut of the cat flea after infection with *Serratia marcescens*, another Gram-negative member of the Enterobacteriaceae [304]. Furthermore, a mutant strain of *Y. pestis* sensitive to reactive oxygen species colonizes fleas with a similar intensity to the wild type, but the percentage of individual fleas of two flea species, *O. montana* and *X. cheopis*, that have cleared the infection seven days p.i. is significantly higher compared to wild-type infected fleas [127]. Therefore, the ability to resist reactive oxygen species, while helpful, is not a prerequisite for *Y. pestis* survival and the maintenance of flea infections. Furthermore, not all results are conclusive, or species-specific differences may exist, as no strong induction of reactive oxygen species was observed in *X. cheopis* following *Y. pestis* infection [83].

In summary, fleas respond to pathogens such as *Y. pestis* with humoral defense mechanisms. However, it is not yet known to what extent these mechanisms contribute to the elimination of the pathogen in a proportion of infected fleas, as has been observed in many experimental studies [51,127,244]. *Y. pestis* has evolved various mechanisms to evade the immune response of fleas.

### 7.2. Effects of Bacteria on the Intestine, Behavior and Fitnes of the Fleas

When investigating the response of the cat flea *C. felis* to ingestion of blood and other microbiota during *Bartonella* infection at the proteome level, the abundance of various proteins differs between fleas fed with uninfected blood and those fed on *Bartonella*-infected hosts [165]. The results are similar to those from parasite-vector systems: approximately 1% (>100 transcripts) of sand fly transcripts are differentially expressed in response to *Leishmania* infection [305]. In tsetse flies, >10 proteins are found in the midgut, which are particularly abundant in trypanosome-infected tsetse flies [306]. However, the parasite’s genetic alterations are significantly more extensive: 25 to 40% of genes are differentially expressed after arrival in the tsetse fly midgut [153]. Interestingly, these changes may be based on the perception of invader components, e.g., through pathogen-associated molecular patterns [168], but also on direct manipulation of the vector by yet-to-be-identified effector molecules of the parasite. In response to high numbers of *Y. pestis*, the rat flea *X. cheopis* not only upregulates the transcription of immune genes (see Section 7.1), but also genes encoding functions of intestinal peristalsis, digestion and peritrophin production [83].

Furthermore, the *Yersinia*-flea system is one of the best-known examples of parasitogenic/pathogenic alterations in vector behavior that increase the likelihood of multi-host infection [250,307,308]. Due to foregut blockage, which is more pronounced in more virulent *Yersinia* strains [309], the flea is unable to ingest blood or only ingests small amounts of blood. Increasing hunger prompts the flea to make repeated attempts to feed or change hosts—all potential sources of transmission.

Regarding the effects of bacteria on flea fitness, neither *R. felis* nor *R. typhi* reduce the fitness of cat and rat fleas, respectively [8,9]. This is also seen in *Bartonella* infections of *Xenopsylla ramesis* [204]. However, the replication kinetics and density of *Y. pestis* influence the fitness of flea vectors. Fleas transmit *Y. pestis* at different temperatures, but at 10 °C, no transmission occurs in the early phase, and during the first two weeks, infected rat fleas have the highest *Yersinia* load, and flea survival is significantly reduced [256,257]. The dominant growth of *Yersinia* at this temperature could be due to reduced competition. *Y. pestis* grows moderately to slowly and is sometimes overgrown by other bacteria under elevated conditions. The reduced flea survival rate could be due to increased blockage. In infections of *Xenopsylla skrjabini* with *Y. pestis*, higher infection inocula correlate with a shortened flea lifespan [264]. Completely blocked fleas can no longer feed blood and starve to death within a few days [62]. Survival time after blockage also depends on the species, averaging two days for *X. cheopis* and four to seven days for *O. montana* [51,150,196,238]. This effect is also observed in *C. felis* when not daily fed. Blockage can occur, and the fleas die within two to four days [128]. In contrast, infected but unblocked fleas do not exhibit increased mortality and can survive for up to 100 days [77,126].

The effects of *Y. pestis* on fleas can be summarized as follows: The bacterium induces humoral immunity but resists it. Blood ingestion is severely impaired by the masses in the foregut, reducing or inhibiting blood ingestion. The hungry fleas repeatedly attempt to ingest blood, increasing the likelihood of *Y. pestis* transmission. The reduced blood uptake due to the blockage in the foregut impairs flea survival.

## 8. Comments for Future Research

Working with diseases transmitted by blood-feeding insects is often complicated and is made more difficult, for example, by government feeding regulations of blood sucking vectors. Artificial feeding systems, while recommended for many vectors, are only accepted by a small portion of the insect population originating from wild populations [64,178]. Since often only a small proportion of insects reproduce after transfer from nature to the laboratory, scientists should ask whether the results reflect the natural scenario. Many flea species reject artificial feeding systems. The cat flea, on the other hand, has accepted them and is maintained this way for almost 40 years [310]. The main vector, *X. cheopis*, is also more easily adapted to laboratory breeding, and some colonies have been maintained for over 20 years [196]. The higher transmission capacity after flea feeding on living hosts compared to artificial systems [264], including pathogens, indicates methodological difficulties. Furthermore, the orientation of fleas to the hosts cannot be studied in such a population.

Similarly to fleas, longer passages of bacteria may result in attenuation. It is known from other microorganisms, including *B. henselae*, that in vitro passages can significantly influence growth kinetics and virulence [311,312]. Since bacteria and vectors coevolve over the long term, strains of the vector and *Y. pestis* should originate from the same location. After a short cultivation period, the bacteria can be stored at −80 °C, and aliquots can be used for new experiments. Even then, the pathogen must be cultured before use in infection/transmission experiments, sometimes for two or more passages. This represents an artificial, non-selective growth phase in which the pathogen is not exposed to its natural blood environment with its nutrients and immune effectors. Furthermore, stock samples may have to be renewed after years or decades, which would extend in vitro generation times.

Fleas should be fed on live hosts, and the bacterium should be cyclically transferred between flea and mammal. Such studies reflect the natural scenario but are logistically more complex and carry a higher biosecurity risk. Furthermore, fleas collected in the field are often stressed, which increases mortality rates and can affect interpretation. Short-term adaptation through daily blood feeding can improve survival rates but is still associated with high mortality rates of uninfected control fleas [228]. In addition, even laboratory strains kept for many years such as *Oropsylla* sp. may show increased mortality due to stress, which in turn affects the experimental results. Optimizing the breeding, e.g., by use of sawdust, can greatly reduce mortality [196].

Many knowledge gaps regarding flea-*Yersinia* interactions have been addressed in other reviews, including the complex molecular interactions during flea passage, the physiological state of *Yersinia* during long-term infection, the genetic and molecular regulation of biofilm production and its release mechanism to facilitate transmission, the response of fleas to oral ingestion of *Y. pestis*, and factors influencing vector competence [34,61,62]. The need for a more standardized study design was recognized due to conflicting results [196]. Results from studies on the efficiency of the early phase of transmission and foregut obstruction were often obtained in studies that differed in many parameters (flea and *Yersinia* strains, inoculum concentration, blood source, natural versus artificial infection, sampling times, methodology used, etc.), so more controlled studies are needed [61,196]. Some methodological issues associated with laboratory flea breeding may have led to underestimation of blockage rates in previous studies, as mortality rates of fleas, including *X. cheopis* and *O. montana*, are increased by stress [196]. However, a certain proportion (up to half) of fleas (with variations in species and blood source) can overcome infection within the first week despite an infection inoculum of >10^8^ CFU/mL [77]. Interestingly, studies on infection rates of tsetse flies by *Trypanosoma brucei* ssp. were also inconsistent until the age of the tsetse flies was recognized as a major factor. While newly emerged flies possess juvenile, thin peritrophic membranes and are highly susceptible to midgut infection, older flies exhibit thickened peritrophic membranes and become more resistant to midgut infection [153]. Although fleas do not form these membranes, other age-related factors can influence the course of the infection, e.g., the accumulation of factors such as peritrophins or toxic compounds through repeated blood ingestion or the further maturation of adult tissue. It would be worthwhile to examine the age of the fleas in future studies.

Furthermore, standardization could be extended to specific methods. For example, information on recovery rates of CFU tests (viable cell counts) is sometimes provided in publications but is rare. Viable cell counts can vary considerably within and between laboratories and depend on many factors, such as media, supplements, sample matrix effects, dilution accuracy, inoculation techniques, incubation time, etc. Improving analytical control by establishing internal standards with defined concentrations, spiked onto an authentic matrix (e.g., whole-body homogenates or specific organs), stored as homogeneous aliquots at −80 °C, and subsequently establishing acceptance criteria for experimental recovery can provide further tools to understand/control potential differences within laboratories and can also be used for interlaboratory studies to demonstrate comparability between different laboratory groups. This concept can also be applied to other analytical methods in experimental studies. In addition, the experimental conditions, including the duration of blood feeding, were identified as an important factor, as *Y. pestis* can multiply in the blood after it has been excreted by the fleas during the experiment. Depending on the feeding time, this can lead to a multiple overestimation of the CFU [128].

In *Bartonella* research, the various forms of bartonellosis cannot be perfectly replicated in animal models. One factor is the high host specificity of some *Bartonella* species [16,19]. This leads to some animals showing no pathological findings despite bacteremia, which in turn complicates the study of the course of infection and disease progression. All currently used animal models, including various mouse strains and dogs, have certain limitations regarding their representativeness of *Bartonella* infections in humans [19].

The situation is even more complicated when researching *Rickettsia* sp. Studies to cultivate viable *Rickettsia* sp. from blood and various tissues of wild animals, even in seropositive animals, have failed. Most studies rely on the detection of rickettsial DNA to demonstrate infection or the presence of infection. Furthermore, there is no suitable animal model, as most animals do not develop clinical symptoms [11], and there is not even a suitable in vitro flea cell line to study the course and dynamics of cell infection [64,174]. Therefore, there is a need to find a suitable animal model and to produce viable *Rickettsia* sp. in culture (e.g., by using susceptible cell lines and endpoint dilution assays to determine the infectious dose) rather than relying on genome copies, which may not correlate with the number of viable cells.

Regarding the development of *Y. pestis* in the flea, it is not yet known whether specialization of biofilm-forming colonies occurs as in *P. aeruginosa* (see Section 6.2). Investigations into the presence of peritrophins on the spines of the proventriculus and their possible involvement or interaction in the colonization and/or stabilization of the pathogen biomass around the cuticle of the proventriculus spines should at least use bacteria from different biovars. These should also be considered in biochemical studies.

The microbiota of fleas should also be studied using field populations, as demonstrated by the high infection rates with *Wolbachia* in laboratory colonies (see Section 4.2.3). Enumerating specific commensals at different time points after blood ingestion and in different gut regions is necessary, as molecular analyses based on transcripts may not correlate with their viability. Laboratory breeding is also known to alter the microbiota of fleas [186], and this effect is likely more pronounced when fleas are maintained in the laboratory for years or decades. Typically, flea colonies that have been reared in the laboratory for more than 10 years are used [196]. Since the interactions between microbiota and *Y. pestis* are largely unknown, this may influence growth kinetics and the outcomes of infection experiments. Considering the interactions of *Y. pestis* with flea immunity, the pathogen-associated patterns of *Y. pestis* are not yet fully characterized [83,299]. Antimicrobial activities in different regions of the intestinal tract remain unclear, particularly the factors of *Y. pestis* that affect other bacteria.

## 9. Conclusions

Only two of the most historically significant infectious diseases—smallpox and rinderpest—have been eradicated to date [313], and neither of these have reservoir animals or is transmitted by vectors. Since vector-borne pathogens/parasites persist despite extensive eradication initiatives or control measures [314], the eradication of plague appears illusory. Therefore, it is recommended to focus on control measures [315,316]. As with other vector-borne diseases, the control of flea-transmitted diseases such as rickettsiosis, bartonellosis and plague requires an integrated approach. Plague has had devastating impacts on human health for two millennia. However, since the mid-20th century, human cases or outbreaks have been rare and predominantly localized in confined areas [32,41]. Global warming may expand the geographic range of fleas, thereby increasing the risk areas for flea diseases [317]. Consequently, ongoing global climate change poses a significant risk for plague resurgence [90,318]. The pathogens must be fully understood to optimize existing prevention strategies and to prevent a resurgence of plague pandemics. The interactions between the pathogens and the vector insects must also be thoroughly investigated. Advanced molecular genetic techniques could offer an approach to interrupting the transmission of *Y. pestis*.

## Figures and Tables

**Figure 1 microorganisms-13-02619-f001:**
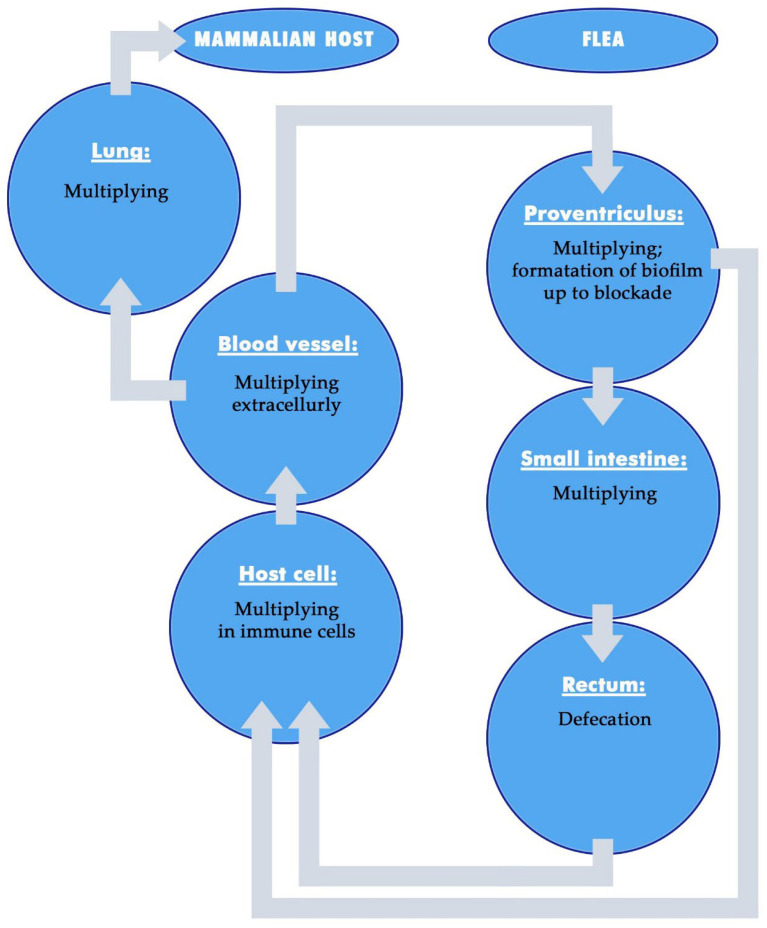
Developmental cycle of *Yersinia pestis* in mammals and fleas.

**Figure 2 microorganisms-13-02619-f002:**
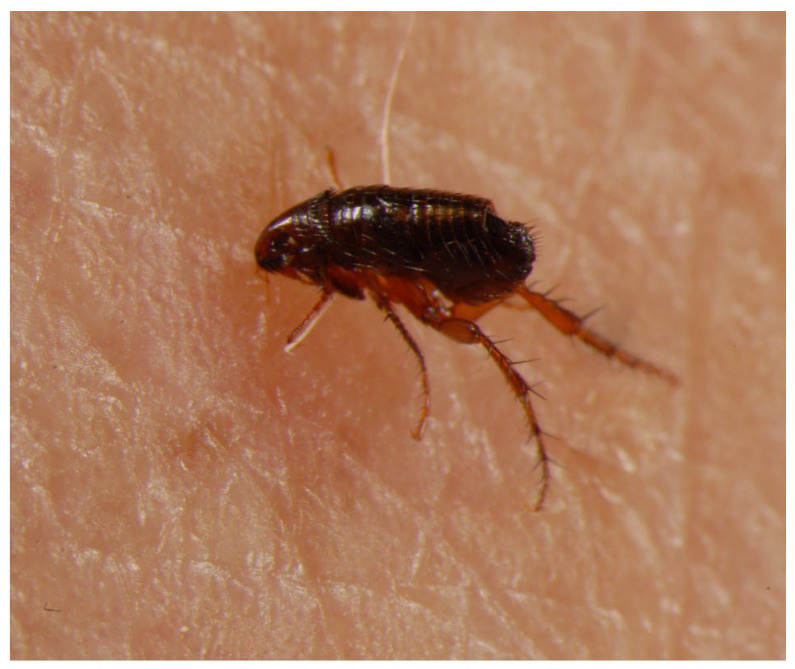
Cat flea *Ctenocephalides felis* during blood ingestion. (Photo: R. Pospischil).

## Data Availability

No new data were created or analyzed in this study. Data sharing is not applicable to this article.

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
