# Peer review of "Interaction of Bacteria and Fleas, Focusing on the Plague Bacterium—A Review"

_microorganisms, 2025, doi:10.3390/microorganisms13112619_

Round 1
Reviewer 1 Report
Comments and Suggestions for Authors
This is a very thoughtful and comprehensive review. I recommend checking some references and take them critically.
Lines 36-37: “…the incidence of human cases in the USA declined sharply in the mid-19th century…” In the mid-19th century, the causal agents of infectious diseases, including Rickettsia species, were unknown; the germ theory of disease itself was not widely accepted until the late 19th century. Therefore, any fluctuations in flea-borne rickettsial diseases during that period would not have been recognized or documented as such.
Lines 43-44: “Several members of the genus Bartonella including B. henselae and B. quintana are zoonotic.” Ironically, B. quintana is only Bartonella spp., which is NOT a zoonotic agent. According to the common definition, a zoonotic pathogen can be transmitted from animals to humans. The only vertebrate host for B. quintana's pathogenic strain is humans. Though there are other strains/subspecies of B. quintana in nonhuman primates, there is no report of human infection caused by these strains. There are two other Bartonella spp. (B. bacilliformis and B. tamiae), for which no animal host has not been detected, but is potentially expected. All the other Bartonella spp. are found in animals and some of them are claimed to cause human diseases, but not B. quintana, which doesn’t require an animal reservoir.
Lines 151-152: “Cat scratch disease usual affects immunocompromised humans.” Most CSD cases are identified among children and immunocompetent cat owners. There are some Bartonella infections, such as bacillary angiomatosis, that do indeed affect immunocompromised individuals.
Line 209: “… flagellated Rickettsia and bartonella are motile.” Most Rickettsia spp., including the well-known pathogens like Rickettsia typhi, are non-flagellated and non-motile. Some species of Bartonella are motile due to flagella, with Bartonella bacilliformis being a well-studied example. While some species have flagella, others, like Bartonella henselae, are non-motile and use other mechanisms to interact with host cells.
Line 276: “Y. enterocolitica… is mainly associated with pigs.” Though Y. enterocolitica could infect pigs, this bacterium has a wide range of animal hosts, including cattle, sheep, goats, dogs, cats, horses, birds, fish, and various rodent species.
Lines 327-328: “Fleas … do not occur above 1,500 feet (450 meters) in altitude and avoid areas with low humidity.” That is not true. In the Altai Mountains, fleas Ctenophthalmus golovi and Amalaraeus penicilliger have been recorded on small mammals at elevations exceeding 2,500–3,000 meters (Medvedev, 1996). In Caucasus and Transcaucasia (Armenia, Georgia), fleas parasitize mice and voles in alpine meadows above 2,000–2,800 meters. In Rocky Mountains, Oropsylla fleas are commonly found on prairie dogs and ground squirrels at elevations of 2,000–3,500 meters. In Sierra Nevada and Great Basin, fleas on chipmunks and marmots occur at 2,400–3,000 meters. On the Tibetan Plateau and Himalayas, fleas such as Neopsylla bidentatiformis and Frontopsylla luculenta, parasitize plateau pikas and voles at elevations reaching 4,000–5,000 meters. In Andes Mountains, fleas parasitizing rodents and guinea pigs have been reported at elevations of 3,500–4,500 meters. Moreover, there are fleas in very dry deserts with a very low level of humility and carry Y. pestis causing human cases.
Lines 335-336: “The flea infestation rate in companion animals … typically ranged 10% and 40%.” Though these high rates are common, the flea infestation rate is more often recorded below 10%. I recommend rephrasing this statement.
Author Response
See file

Reviewer 2 Report
Comments and Suggestions for Authors
The abstract does not adequately capture the manuscript's content. It is limited to the description of very narrow aspects and does not provide a true overview of its contents. Consequently, it must be revised to include other aspects addressed in the text.
The text is plagued with inaccuracies:"
Line 173: “R. felis is highly adapted to the booklouse and is fully maintained by vertical transmission. In addition to the aforementioned host-vector switching, the infection may also be maintained horizontally [56,57].
The booklouse (Liposcelis bostrychophila) is a non-hematophagous (non-blood-feeding) reservoir and vector of Rickettsia felis. This finding is crucial because it helps to explain how people can contract this febrile illness even without contact with fleas or domestic animals. Referenciado en “Behar, A., McCormick, L. J., & Perlman, S. J. (2010). Rickettsia felis infection in a common household insect pest, Liposcelis bostrychophila (Psocoptera: Liposcelidae). Applied and environmental microbiology, 76(7), 2280-2285” and “Mediannikov, O., Bechah, Y., Amanzougaghene, N., Lepidi, H., Bassene, H., Sambou, M., ... & Raoult, D. (2022). Booklice Liposcelis bostrychophila naturally infected by Rickettsia felis cause fever and experimental pneumonia in mammals. The Journal of Infectious Diseases, 226(6), 1075-1083”.
There are also some errors that must be corrected:
Line 208: “These three genera all belong to Proteobacteria, Gram-negative coccobacilli, of which the flagellated Rickettsia sp. and Bartonella sp. are motile.
Bacteria of the genus Rickettsia do not have flagella. They are generally considered non-motile in the traditional sense, as they cannot "swim" on their own in a liquid medium. However, Rickettsia is well-known for its ability to move inside the host cells it infects. To do this, it doesn't use flagella, but rather manipulates the host cell's cytoskeleton: it uses a protein called actin to create a kind of "tail" that propels it through the cytoplasm.
Attention must also be paid to the style; for example, scientific names do not appear in italics.
Author Response
see file
